# A PSII photosynthetic control is activated in anoxic cultures of green algae following illumination

Yuval Milrad[1], Valéria Nagy[2], Tamar Elman[1], Maria Fadeeva[3], Szilvia Z. Tóth[2] & Iftach Yacoby [1✉]

Photosynthetic hydrogen production from microalgae is considered to have potential as a renewable energy source. Yet, the process has two main limitations holding it back from scaling up; (i) electron loss to competing processes, mainly carbon fixation and (ii) sensitivity to $O_2$ which diminishes the expression and the activity of the hydrogenase enzyme catalyzing $H_2$ production. Here we report a third, hitherto unknown challenge: We found that under anoxia, a slow-down switch is activated in photosystem II (PSII), diminishing the maximal photosynthetic productivity by three-fold. Using purified PSII and applying in vivo spectroscopic and mass spectrometric techniques on *Chlamydomonas reinhardtii* cultures, we show that this switch is activated under anoxia, within 10 s of illumination. Furthermore, we show that the recovery to the initial rate takes place following 15 min of dark anoxia, and propose a mechanism in which, modulation in electron transfer at the acceptor site of PSII diminishes its output. Such insights into the mechanism broaden our understanding of anoxic photosynthesis and its regulation in green algae and inspire new strategies to improve bio-energy yields.

[1] School of Plant Sciences and Food Security, The George S. Wise Faculty of Life Sciences, Tel Aviv University, Ramat Aviv, Tel Aviv 69978, Israel. [2] Institute of Plant Biology, Biological Research Centre, Szeged, Temesvári krt. 62, H-6726 Szeged, Hungary. [3] Department of Biochemistry, The George S. Wise Faculty of Life Sciences, Tel Aviv University, Ramat Aviv, Tel Aviv 69978, Israel. ✉email: iftachy@tauex.tau.ac.il

Photosynthetic electron flow is crucial for the development of complex life on earth, as in this process, sunlight is captured as a primary energy source for most living organisms. This process is mostly renowned for its role in $O_2$ evolution by photosystem II (PSII) of cyanobacteria, algae, and plants[1,2]. However, harvesting sunlight, although very efficient as a primal energy source, does not come without challenges. One of the main issues that plants deal with is the instability and inconsistency of the irradiance levels. To overcome these problems, plants have evolved a sophisticated network of regulatory processes that tune the efficiency of the photosynthetic apparatus. The process of electron flow is constantly regulated to bypass barriers according to the availability of metabolites of the photosynthetic apparatus. Redox poise and sub-localization of complexes in the thylakoid membrane were both postulated to pose "photosynthetic control" and maintain electron flux to fit the system's capacity[3,4]. Green microalgae, which are continuously subjected to environmental changes, have also evolved several regulatory mechanisms to cope with rapid changes in light quality and intensity[5]. These processes enable cells to cope with fast transitions from darkness as they increase the amount of available downstream products of both photosystems, and thus alleviate acceptor side limitations. However, when such a transition is taking place under anaerobiosis, due to the cells' respiration or external $O_2$ scavenging, $H_2$ evolution by hydrogenase, which is otherwise prone to inactivation by $O_2$[6], becomes the only effective valve to cope with an excess of energy upon such sudden light exposures[7–9].

The production of $H_2$ from green algae attracts much research, as it is considered to be a potential renewable energy source. Recently, a possible breakthrough towards its scalability was achieved, as a prolonged ambient $H_2$ production was demonstrated[10,11]. Accordingly, it was shown that the two main challenges that have been holding back scalable $H_2$ production from green algae (inactivation by $O_2$ damage and competition with $CO_2$ fixation[12,13]) are resolvable. Indeed, it was previously shown that challenging the cells with fluctuating light, ranging minutes or below, can limit $O_2$ to low levels and thus improve the sustainability of $H_2$ production[7,14,15]. However, such attempts resolved another, yet unidentified barrier for $H_2$ evolution. It was shown that the initial exposure to light, following dark anaerobic incubation, triggers a fast flux of electrons, as reported by high rates of $H_2$ evolution. In contrast, successive exposures, result in a 3-fold decrease in $H_2$ accumulation, regardless to the number of dark-light cycles[7]. To date, the mechanism responsible for this dramatic decline remains elusive. In this work, we explored the origins of this massive decrease via the assessments of global and local electron fluxes in intact algal cultures, and purified PSII complexes. We recorded and integrated the electron transport processes from the oxygen evolving center (OEC) in PSII, to processes downstream of PSI. Our results suggest that the redox activated "photosynthetic control", which is responsible for a slowdown in Cyt$b_6f$ activity, generates acceptor limitations on PSII, which alters its inner electron flow mechanism, and cause a massive reduction in the effective electron output. This downregulation, possibly involving the photoreduction of $O_2$ at the acceptor site of PSII and possibly an alternate conformation of its acceptor site residue arrangement, is later translated into a remarkable decrease in $H_2$ production.

## Results

### Following a short light exposure, photosynthetic productivity is decreased by 3-fold.

Dark anoxic cultures of green microalgae immediately emit $H_2$ at high rates upon exposure to light. To examine its kinetics, *C. reinhardtii* cells were cultivated in mixotrophic medium and incubated under dark anoxia for an hour. As was previously described[7], following incubation, we exposed the cells to light (370 µE m$^{-2}$ s$^{-1}$) for 2 min, and the concentrations of evolved $H_2$ and $O_2$ were monitored using a membrane inlet mass spectrometer (MIMS)[16] (Fig. 1a, b). As expected, following an initial burst, $H_2$ evolution rate declined rapidly to a complete cessation. As $H_2$ accumulation plateaued, the light was turned off. Importantly, tracking $O_2$ accumulation enabled us to keep the cells under darkness for enough time to completely respire $O_2$ (3–5 min), hence maintain anoxia. We then illuminated the cells again, for 2 min and observed that $H_2$ evolution resumed. These light/dark fluctuations (cycles) were repeated four times, and although $H_2$ was evolved in all successive light exposures, we observed a ~3-fold global decrease in $H_2$ accumulation rates in comparison with the first exposure to light (Fig. 1a—dashed vs. solid). Interestingly, we also observed that the initial exposure triggered an immediate linear increase in net $O_2$ evolution, in contrast to successive exposures in which an exponential increase can be seen (Fig. 1b—dashed vs. solid). To assess whether these differences are limited only to mixotrophic growth, we conducted a similar test, using cells which were cultivated under autotrophic conditions (Supplementary Figure 1). Notably, the $H_2$ production rates were lower in photoautotrophically grown cultures, as we previously reported[7]. Since these conditions stimulate higher rates of $O_2$ evolution, the accumulated $O_2$ increase the competitive inhibition of hydrogenase activity (by reactions such as 'Mehler-like', Mehler and others[17]). Therefore, we conducted these measurements in the presence or the absence of $O_2$ scavengers (Glucose oxidase [GOx], supplied with glucose and catalase). The addition of such scavengers slightly increased the amount of the accumulated $H_2$, from $10.7 \pm 1.7$ µM $H_2$ in their absence to $14.4 \pm 1.3$ µM $H_2$ in their presence (roughly a half of the value accumulated under mixotrophic adapted cells, $32.6 \pm 2.5$ µM $H_2$). In addition, as observed for the mixotrophic conditions, we observed a stark decline in $H_2$ production rates, between the first and successive light exposures (for both treatments), which stood at $50 \pm 1\%$ and $65 \pm 6\%$ inhibition, in the absence or presence of $O_2$ scavengers, respectively.

### Linear electron flow is downregulated by light exposure.

Net $O_2$ measurements are not sufficient to correctly estimate the efficacy of PSII because they can only reflect its gross production at some level. Therefore, we also tested alterations in photosynthetic efficacy by measuring chlorophyll (Chl) *a* fluorescence[18,19] (Fig. 1c). The kinetics of Chl *a* fluorescence, following dark anoxia features two peaks[20]. The first peak is reached immediately following light exposure, as all $Q_B$ sites are occupied with plastoquinol[21]. The second peak is reached several seconds later due to either; (i) the activation of downstream processes, specifically $CO_2$ fixation by the CBB cycle[22], (ii) conformational changes in PSII[19,23], or (iii) distribution of LHCII antenna complexes (state transition)[24]. Following these peaks, the fluorescence signal reaches a steady state. As expected, we observed that at the initial light exposure, the signal increased for a duration of 30 s (dashed in Fig. 1c)[25]. In contrast, successive illuminations triggered an immediate sharp increase, in the timeframe of the first peak of the initial exposure and featured no additional peak. It should be noted, however, that following 60 s of illumination all fluorescence traces equalized, and steadily declined in the same manner. To assess whether heat dissipation or state transition were altered, we exposed the cells to a saturating pulse, as the traces reached steady state (see red arrows in Fig. 1c) and determined the maximal fluorescence (Fm'). We observed no

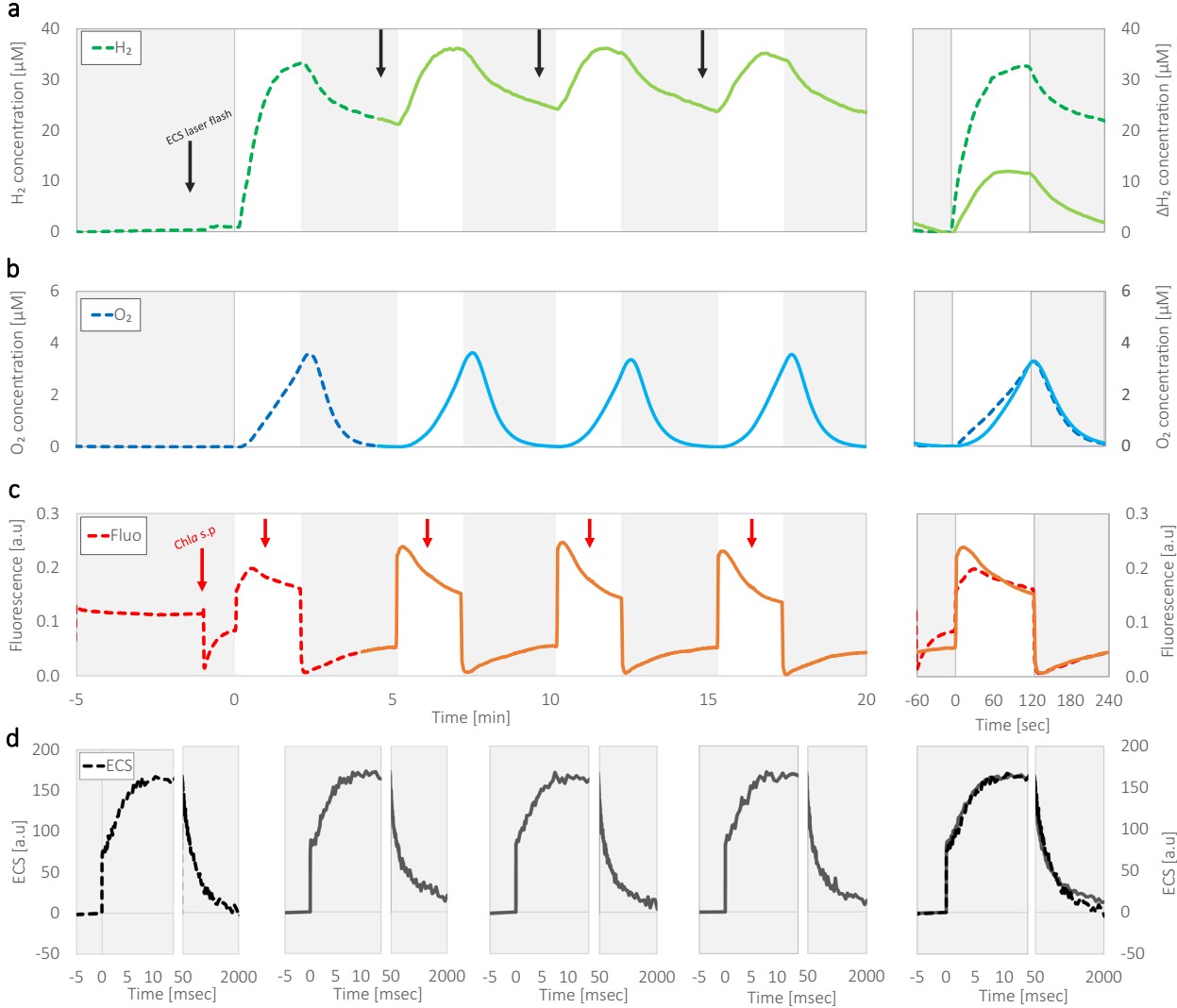

**Fig. 1 Hydrogen production following light and dark fluctuations.** Mixotrophic *C. reinhardtii* wild-type cultures (strain CC124) were incubated for an hour under dark anaerobiosis, after which they were challenged with light fluctuations of 2 min under illumination (at an irradiance of 370 μmol photons m$^{-2}$ s$^{-1}$, white background), followed by 3 min of darkness (gray background). Shown are the differences between the initial light exposure (dashed) and the average of the successive exposures (solid). H$_2$ (**a**) and O$_2$ (**b**) concentrations were measured using MIMS, and Chl *a* fluorescence was measured simultaneously using PAM (**c**). During illuminations, the cells were exposed to saturating pulses (marked with a red arrow), to assess maximal fluorescence intensity. The same protocol was used to assess changes in electrochromic shifts (at 520–546 nm) by a JTS (**d**), for which the cells were exposed to a laser flash of 30 s prior to each light exposure (see black arrows in panel **a**). The right graphs in each panel show a comparison between the initial (dashed) and the average of all three successive (solid) exposures, relative to their state at light onset, or laser flash (time, 0). Each curve represents the averaged result of at least 3 biological repetitions.

significant changes between light exposures (Supplementary Figure 2), indicating minimal changes if any in the magnitude of non-photochemical quenching, nor a decrease of PSII efficiency due to state transition as the cells reach steady state. To verify that indeed no shifts in the location of LHCII antennas took place, we sampled cells directly from the experiment's cuvette and examined their fluorescence spectra at 77 °K (Supplementary Figure 3). We observed that the peaks, which are observed at ~680 nm and ~710 nm, show no alterations between the first and consecutive light pulses, which indicates that the time of light exposure (2 min) is not sufficient to generate state transition[26], in accordance with previous research[9].

To verify that the changes are not originating from alterations in the overall efficiency of the photosystems, we examined differences in electrochromic shifts (ECS) between light exposures, by tracking changes in absorbance at 520 and 546 nm[27] (Fig. 1d, see also; black arrows in Fig. 1a). When cells are exposed to a single turnover laser flash their absorbance is changed via a 3-phase shift[27]. The initial increase in the ECS signal (termed "phase a") lasts less than a millisecond and is a product of charge separations taking place in both photosystems, hence decreased signal can report on the diminishment of PSII and PSI activity. During the second phase (termed "phase b"), which usually lasts up to 10 ms[28], ECS signal increases due to a build-up of membrane potential, mainly via Cyt$b_6f$. In the final phase (termed "phase c"), the signal decreases exponentially due to the dissipation of the membrane potential via ATPase activity. Here we observed that all flashes, which were given before each light exposure (see black arrows in Fig. 1a), triggered an identical shift, indicating that the redox state at light onset was similar between light fluctuations (Fig. 1d).

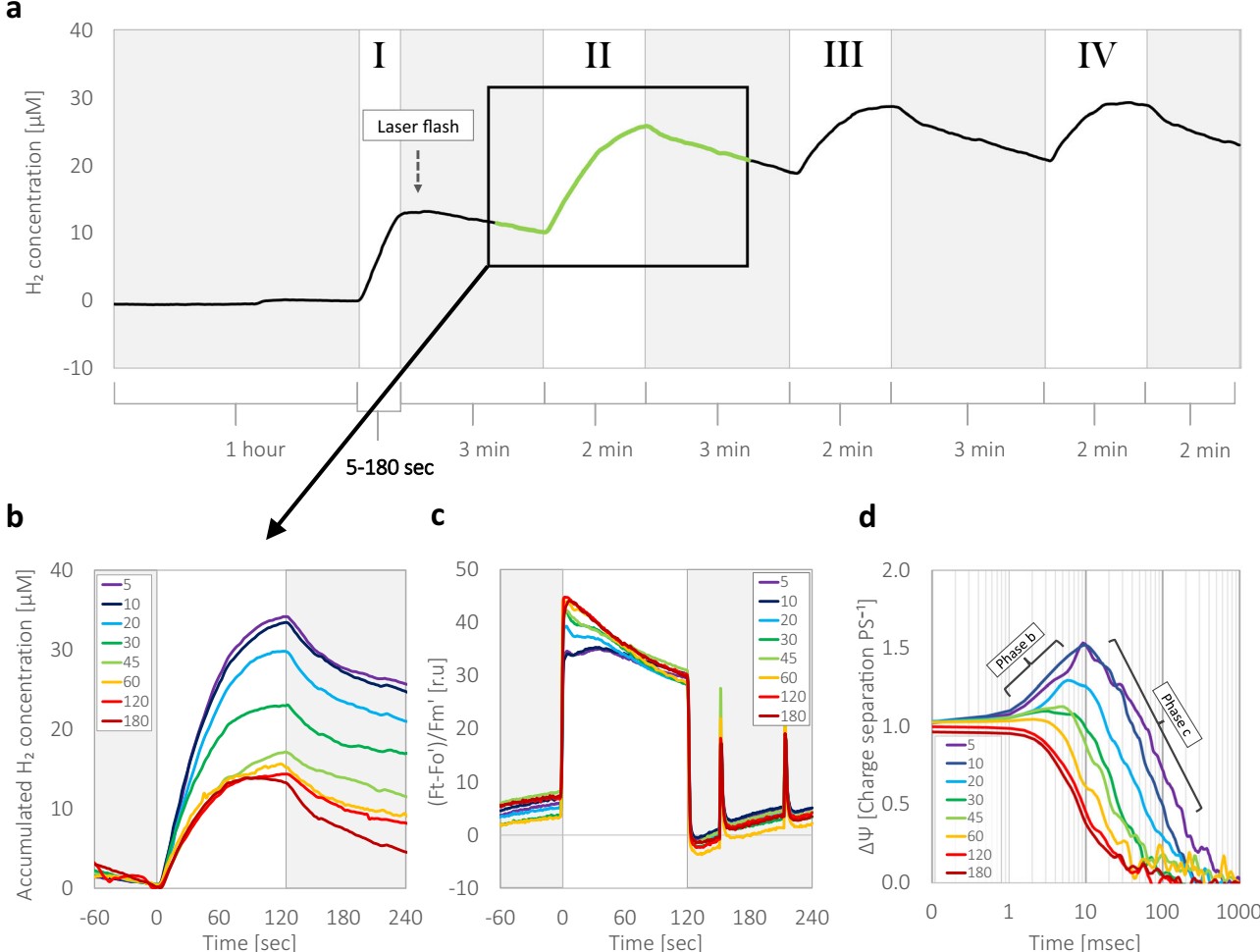

**Fig. 2 Following dark anoxia, light exposure longer than 10 s gradually slows down electron output from PSII. a** Mixotrophic *C. reinhardtii* wild-type strain CC124 cells were tested for $H_2$ evolution in a MIMS. The cells were incubated for an hour under dark anaerobiosis, after which they were illuminated (370 µE m$^{-2}$ s$^{-1}$) for a duration of either 5, 10, 20, 30, 45, 60, 120, or 180 s (Exposure I, see bolted time scale). Presented here is the trace, which was measured by exposing the cells for a duration of 45 s in exposure I. Following the initial light exposure, the cells were kept in a darkness for 3 min (gray background) and illuminated again for 2 min, three times (Exposures II, III, and IV, white background). To compare the effects of the duration of exposure I on photosynthetic regulation, results which were measured during exposure II were plotted against the accumulated concentration of $H_2$ (**b**) and photosynthetic efficiency, which was assessed by Chl *a* fluorescence measurements (**c**). Electrochromic shifts were determined by measuring changes in cells' absorbance (520–546 nm). Three seconds following exposure I (see dashed arrow in panel **a**), the cells were exposed to a 5 ns laser flash, and charge separation was measured (**d**). The color index in all panels matches the duration of exposure I (ranging seconds): 5—purple, 10—blue, 20—cyan, 30—green, 45—light green, 60—yellow, 120—red, and 180—dark red. Each experiment was repeated using at least three biological replicates. Error bars indicate standard error (*n* ≥ 3).

**Light activation of photosynthetic regulation mechanisms**. It was previously shown that activation of PSI cyclic electron flow, by continuous exposure to light increases the proton motive force across the thylakoid membrane, which initiate "photosynthetic control"[29], and thus cause a shift in the rate-limiting steps of the photosynthetic apparatus. Photosynthetic control is activated by elevated redox poise, which is dependent on the duration of irradiance. Such regulation was proposed to diminish linear electron flow[3], and subsequently decrease $H_2$ production[30]. Hence, one could suggest that such changes in electron flow should require a sufficient time for a build-up of the redox poise across the thylakoid membrane. To test this hypothesis, following an hour of dark incubation, we exposed the cultures to light, for a varied duration of time, between 5 and 180 s (Fig. 2a). Then, the cells were kept under darkness for 3 min, before they were exposed to light for a second time, for additional 2 min. Two more fluctuations (of 3 min darkness/2 min of illumination,

marked as exposures III and IV) were performed to verify that indeed no further changes occur during the experiment. We then tracked the evolution of $H_2$ (Fig. 2b) and slow kinetics of Chl *a* fluorescence (Fig. 2c) during the second 2 min illumination (see square in Fig. 2a) and plotted the results in accordance with the preceding duration of the initial exposure (meaning exposure I). Our results show that the activation of the photosynthetic regulation mechanism is gradually increasing during the first 30 s of illumination (in accordance with our observation in Fig. 1b).

To gain better understanding on the activation of "photosynthetic control", we monitored the build-up of redox poise, which is associated with it. We exposed the cells to a short laser flash 3 s following the initial light exposure (see black arrow in Fig. 2a), and measured changes in the kinetics of ECS build-up (Fig. 2d). We observed that, although no differences in rapid charge separation at "phase a" are apparent (Supplementary Figure 4), the build-up of the redox potential, seen by the lack of

an apparent rise in the trace at "phase b", is gradual which is in accordance with the decrease in $H_2$ production. Such changes of the redox potential should also decrease the electron flux via Cyt$b_6f$, due to the activation of "photosynthetic control", and therefore gradually pose a donor-side limitation on PSI and an acceptor side limitation on PSII. Such limitation could ultimately result in a decreased rate of linear electron flow, which is also seen as a lower rate of $H_2$ production. It is possible of course that an amplified activity of ATPase, which is observed as an accelerated decline in "phase c", decreases the build-up of membranal charge and therefore diminish the amplitude seen in "phase b"[31]. However, this should again indicate a rapid formation of redox poise, as it is needed for inducing such an accelerated activity of ATPase, and therefore, should also result in "photosynthetic control" generation that would form acceptor limitations on PSII. These results are in line with the notion that under anoxia, the formed redox poise diminishes the rate of linear electron flow. Yet, here we suggest that these traits generate a graver limitation, which is situated within PSII, by an alteration of its working mechanism.

**Diminished activity of isolated PSII complexes under anoxia in the presence of high bicarbonate concentrations.** In order to pin-point the cause of the apparent decrease in linear electron flow, we evaluated the effect of anoxia on the activity of isolated PSII. To that aim, we purified PSII complexes from *C. reinhardtii* wild-type strain CC124 cells, and examined $O_2$ evolution rates, using a Pyroscience FireSting $O_2$ probe[32] (Fig. 3a). We exposed purified PSII complexes to actinic light in the presence of 2,5-dichloro-p-benzoquinone (DCBQ) so they would not face acceptor side limitations on the $Q_B$ site. The complexes were exposed to 10 s of illumination, in which $O_2$ evolution rates

were determined, before the light was turned off for a minute. Then, we exposed them for another 10 s of illumination, and determined the residual activity of the 2nd exposure, in relations to the 1st. Since we examined the effects of anoxia on the activity of PSII, we conducted the experiment under either aerobic or anoxic conditions, which were established by flushing the samples with $N_2$ (Fig. 3a, Aerobic vs. Anoxia, respectively). In addition, to assess the effects of bicarbonate (which was added in the form of NaHCO$_3$, see "Methods"), we conducted the same experiment in its presence or complete absence. The results show that the dark anoxic conditions gravely decrease PSII activity by up to 70%, upon initial light exposures, in line with previous observations[33]. The results also show that the initial exposure to light was not affected by the presence or absence of NaHCO$_3$ (under neither aerobic nor anoxic conditions). It should be noted that the $N_2$ sparging, lowered the bicarbonate concentration from 10 mM to 7.5 mM as measured by MIMS (Supplementary Figure 5). Interestingly, we observed that upon the 2nd illumination, $O_2$ evolution rates were not affected under aerobic conditions, neither in the presence or absence of NaHCO$_3$. However, anoxic conditions triggered a decreased activity of ~45% in the 2nd illumination (with a significance of p.v. = 0.0154 in a student's t-test), only in the presence of NaHCO$_3$.

In our in vivo examinations, we observed that the inhibition of the apparent PSII activity increases under prolonged light exposures (Fig. 2). To test these effects on isolated PSII complexes, we illuminated them to durations of 5, 10, 20, or 30 s (Fig. 3b, purple, blue, cyan and green respectively, coloration match other panels). As before, the complexes were exposed to two illuminations, the rate of $O_2$ evolution was determined for each exposure, and the residual activity ratio was calculated for each exposure duration. Remarkably, under aerobic conditions, no decrease in PSII activity was observed throughout exposures

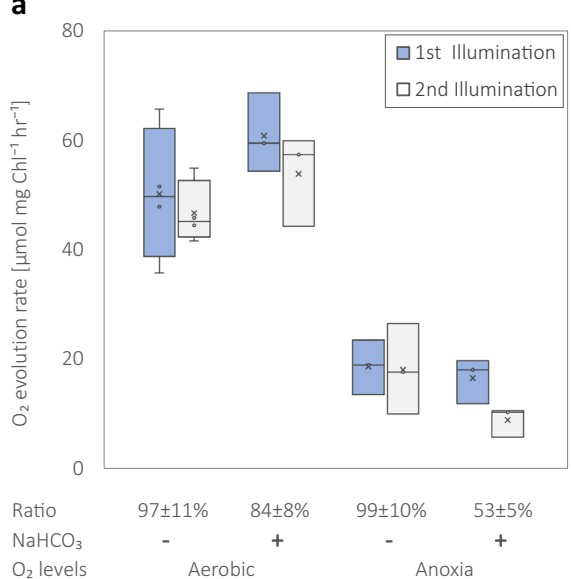

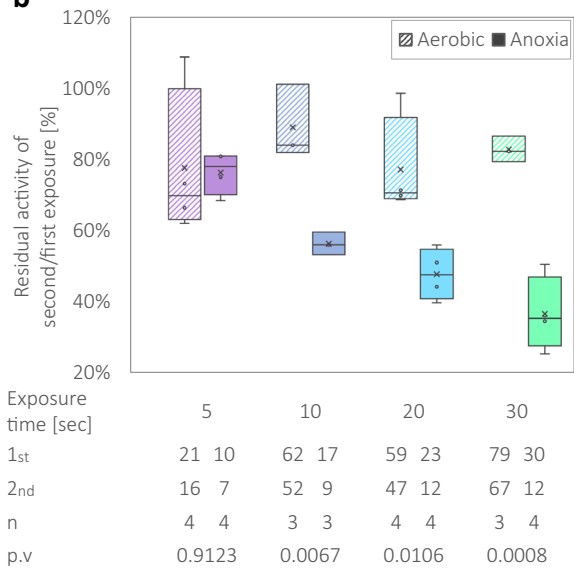

| Exposure time [sec] | 5 | | 10 | | 20 | | 30 | |
|---|---|---|---|---|---|---|---|---|
| 1st | 21 | 10 | 62 | 17 | 59 | 23 | 79 | 30 |
| 2nd | 16 | 7 | 52 | 9 | 47 | 12 | 67 | 12 |
| n | 4 | 4 | 3 | 3 | 4 | 4 | 3 | 4 |
| p.v | 0.9123 | | 0.0067 | | 0.0106 | | 0.0008 | |

| Ratio | 97±11% | 84±8% | 99±10% | 53±5% |
|---|---|---|---|---|
| NaHCO$_3$ | - | + | - | + |
| O$_2$ levels | Aerobic | | Anoxia | |

**Fig. 3 Purified PSII complexes show decreased activity at successive light exposures.** PSII complexes were isolated from *C. reinhardtii* wild-type strain CC124 cells, and their activity was tested by a Pyroscience FireSting $O_2$ probe, (**a**). The complexes were tested in the absence or presence of 10 mM (actual 7.5 mM) NaHCO$_3$ to mimic high carbon conditions, and examined under either aerobic or anoxic conditions. Complexes were exposed twice for 10 s of light, with a dark period of a minute in-between. Shown are the rates of the 1st (blue) and 2nd (white) exposures, for each set of conditions. Also stated are the residual activity rates between each aligned exposure (according to the averaged rate). Complexes in the presence of NaHCO$_3$ were also tested during increased periods of illuminations of 5, 10, 20, or 30 s (purple, blue, cyan, and green, respectively), under either aerobic (striped) or anoxic (solid) conditions in the presence of 10 mM (actual 7.5 mM) NaHCO$_3$ (**b**). Presented are the residual rate of $O_2$, which was accumulated in the second exposure, compared to the initial illumination. The table below presents the rates of the 1st and 2nd exposures for each treatment (µmol $O_2$ mg Chl$^{-1}$ h$^{-1}$) as well as the repetitions number for each result. In addition, the difference between the aerobic and anoxic treatments were analyzed using a student t-test. *P*-values are stated for each couple. Each experiment was repeated using at least three biological replicates. Data is presented in box plots.

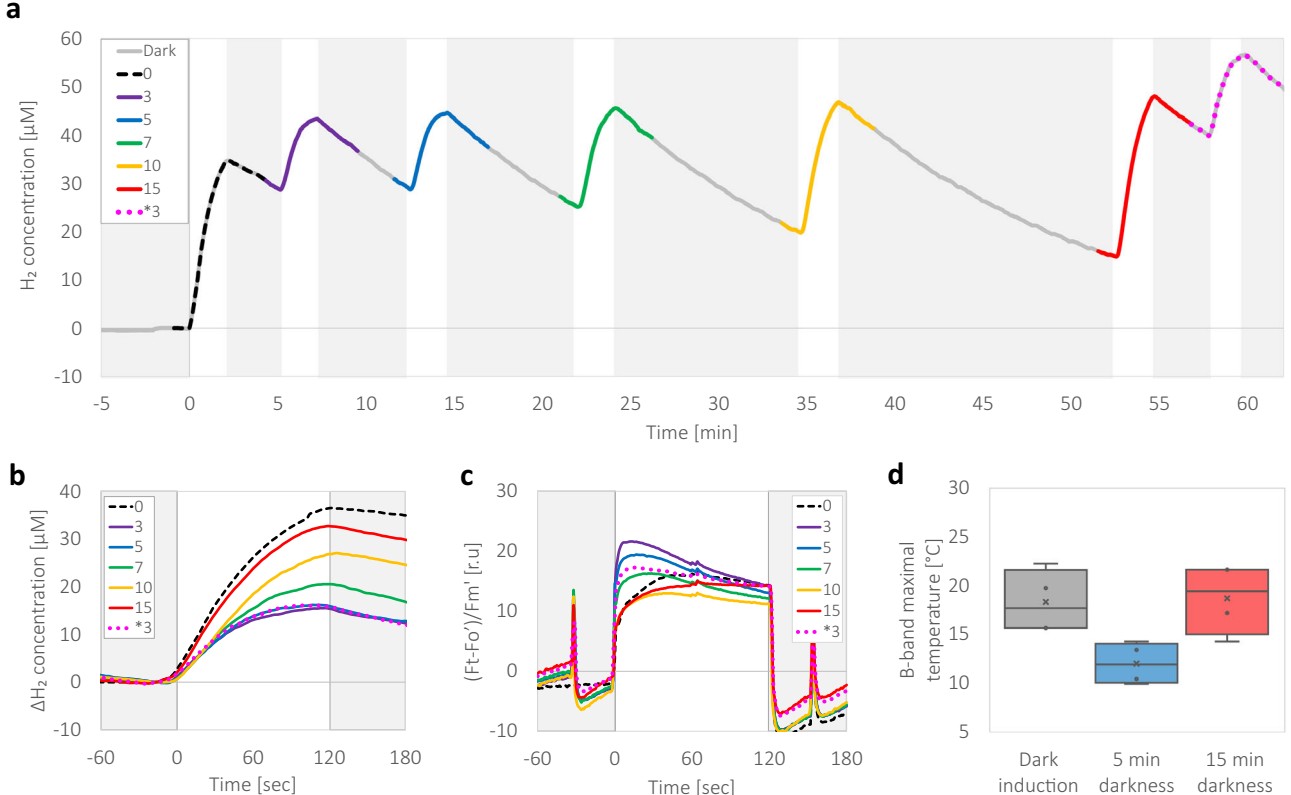

**Fig. 4 A dark anoxic incubation reestablish the initial fast electron flux.** Following an hour of dark incubation in the presence of $O_2$ scavengers (GOx), *C. reinhardtii* wild-type strain CC124 cells were subjected to a series of fixed duration of light exposures (370 µE m$^{-2}$ s$^{-1}$ for 2 min, white background) hatched with an increasing duration of anoxic dark incubations 0–15 min. **a** $H_2$ accumulation as a function of the preceded dark incubation time (gray background) between fixed 2 min of irradiance. The production of $H_2$ in each light exposure was measured by MIMS and the trace is highlighted according to its preceding duration of dark incubation (initial exposure, 0—dashed, 3 min—purple, 5 min—blue, 7 min—green, 10 min—yellow, 15 min—red, additional 3 min—dotted pink, the same color index was used for all the traces in all panels). **b** To assist the comparison between the differences in $H_2$ accumulation traces shown in panel **a**, all the traces measured at each light period are plotted at once using a single fixed time frame. **c** To assess the changes in the PSII photosynthetic efficiency, Chl *a* fluorescence was simultaneously measured. During each illumination, the cells were exposed to a saturating pulse and maximal fluorescence (Fm') was determined. Photosynthetic efficiency was then normalized to Fm'. **d** Thermoluminescence of intact algal cells was measured after 2 single turnover flashes (STF) spaced 1 s apart. Samples were measured following an hour of dark anaerobic incubation (gray). Then, they were illuminated for 2 min followed by a dark relaxation of either 5 (blue) or 15 (red) minutes, and the temperature in which the maximal values for the B-band were detected. The gained values were plotted in box plots. Each experiment was repeated using at least three biological replicates.

and sustained ~85% activity (Fig. 3b, striped columns). In contrast, complexes which were exposed to light under anoxia in the presence of bicarbonate, featured a gradual decrease in their activity in concomitant of increased exposure time durations, and were determined to evolve 63% less $O_2$ during 30 s of illuminations, which is in line with our in vivo observations.

**Dark recovery of the initial electron transfer rate.** Since we observed that light exposure triggered a decrease in linear electron flow, we examined whether such modulations are permanent or could be reversed by prolonged anoxic darkness. To that aim, we gradually increased the duration of darkness in between light exposures, which were set to a constant 2 min of illumination (Fig. 4a). Dark recovery periods were timed to 3, 5, 7, 10, and 15 min, having an additional 3 min of darkness before the last illumination to verify that the observed alterations are indeed due to the duration of darkness, rather than the total time that passed in the experiment (marked with an asterisk; 3*). In addition, to eliminate possible interferences by accumulated $O_2$, we added an $O_2$ scavenger, glucose oxidase (GOx, supplied with glucose and catalase[34]). The results show that the decrease in $H_2$ evolution, following 3 min of darkness is not affected by the absence of $O_2$ (Fig. 4b), and that $H_2$ evolution rate increases as a function of

dark recovery time. We also observed that the cells require at least 15 min of darkness to fully regain the initial $H_2$ evolution rate. Since we previously observed alterations in PSII activity (Fig. 1c), we tracked changes in Chl *a* fluorescence (Fig. 4c) simultaneously with MIMS measurements. Here, we observed the same gradual changes in the early fluorescence rise of ~60 s, which indicates that the activity of PSII was indeed recovered in the dark period of 15 min. To verify that state transition did not lower the fluorescence intensity, we exposed the cells to a saturating pulse 60 s prior and 30 s after each illumination period and observed no differences between paired maximal fluorescence values (Supplementary Figure 6). We could therefore conclude that the potential activity of PSII is not decreased due to photodamage or state transitions.

We then examined alterations in the electron transport within PSII, by assessing the effect of light exposure on intact cells by thermoluminescence (TL). Light emission from a pre-illuminated sample during temperature increase provides valuable information about the charge recombination reactions occurring within PSII (for reviews on the subject, see refs. [35,36]). When intact cells or leaves are exposed to two single turnover flashes, the "B band" appears with a maximum at around 20 to 40 °C, originating from a mixed recombination of $Q_B^-$ with the $S_2$ and $S_3$ states of the

OEC[35]. We measured intact cells following an hour of dark anoxic incubation and observed the maximum B-band intensity at $18.1 \pm 1.5\,°C$ (Fig. 4d, raw traces and the effects of additional flashes on the B-band are presented in (Supplementary Figure 7)). We then exposed the cells to 2 min of light, followed by 5 min of darkness and performed a TL measurement following two flashes. A few minutes of dark adaptation is usually sufficient for a full recovery of the B-band; however, in the anoxic cultures illuminated for 2 min a major decrease in the intensity of the B band was observed, indicating that less charge recombination occurred between the $S_2$ and $S_3$ states of the OEC and $Q_B^-$ or alternatively, recombination occurred via non-radiative pathways[37,38]. In addition, the peak position of the maximum TL intensity was downshifted significantly, to $12.3 \pm 1.0\,°C$ (with a $p$-value of 0.005 in a student's t-test). These changes indicate that the redox equilibrium between $Q_A^-$ and $Q_B^-$ was changed towards $Q_A^-$, facilitating charge recombination between $Q_A^-$ with the donor side. To assess whether indeed these shifts are reversed during longer periods of anoxic darkness, we exposed the cells to a second 2-min period of illumination followed by 15 min of anoxic darkness. We observed that both the intensity and the peak position of the B band ($18.4 \pm 1.7\,°C$) largely recovered following 15 min of dark anoxia. These findings indicate that charge recombination reactions within PSII are altered by light exposure in a reversible manner. The data also suggest that the cause of the apparent decrease in the output of the total electron flow under anaerobiosis can be accounted for changes occurring within PSII.

## Discussion

In nature, green algae are subjected to various environmental changes. Since they are mostly rely on operation of the photosynthetic apparatus, many of these changes are resolved by the cells' ability to rapidly alternate the course of their metabolic electron flow[39,40]. Indeed, the activation or deactivation of electron sinks has been the topic of much research recently, and was shown to hold a crucial role in the ability of the cells to sustain functionality[5,7,8,15]. In this work, we demonstrate that green algae under anoxia have additional regulation mechanisms which have a profound effect on the overall productivity of the photosynthetic apparatus, decreasing its electron output (Fig. 1). This mechanism is activated within 10 s of illumination and results in a massive slowdown of electron flow by 3-fold (Fig. 2). This mechanism is reversible and can be turned off (gaining back the fast electron flow) following a dark anoxic incubation for 15 min (Fig. 4).

Following anaerobic induction, alternative electron flow pathways together with enhanced ATPase activity generate an additional redox potential and ATP supply, $CO_2$ fixation initiates, and a general increase in the electron flux appear[7,14,31]. It was previously shown that the enhanced proton motive force activates "photosynthetic control" in which the rate of electron flux through $Cyt\,b_6 f$ is decreased[3]. Such decrease results in a continuously reduced PQ pool and hinder the activity of PSII, thus generating an excessive pressure on its acceptor side. This situation provides the ground for the hypothesis raised by Cardona et al, stating that: "An acceptor side switch, should it exist, could for example be triggered by the formation of $Q_A^-$ before the $Q_BH_2$ has left the site or formation of $Q_B^-$ before $QH_2$ has cleared the channel and remains in the vicinity of the heme"[41].

During aerobic light exposure, the activation of non-photochemical quenching was shown to diminish the yield of PSII light harvesting[42–45], alleviating the pressure caused by excess light. In this work, which is focused in anoxia we also observed changes in Chl $a$ fluorescence (Fig. 2c), which can be

wrongly interpreted as changes in NPQ. However, it is not the energy dependent qE component of NPQ as that should be relaxed within less than a minute nor state transition which was shown to be uncorrelated (Supplementary Figure 3). In addition, the fluorescence differences do not arise from photoinhibition, as the saturating pulses and the ECS measurements showed no differences between light fluctuations (see Fig. 1c, d and Supplementary Figures 4, 6). In fact, we observed that the acceptor side limitations are dissipated in darkness, in a timeframe shorter than 3 min, as evidenced by ECS measurements (Fig. 1d). Lastly, the TL data (Fig. 4d) pinpoints the origin of this anoxic slowdown mechanism to PSII. We can therefore conclude that the generated acceptor limitations involve intrinsic changes in PSII, diminishing its electron output. This decreased output is translated to an apparent decrease in linear electron flow, seen as a decreased rate of $H_2$ production by the cells.

The remaining question hereafter is, what is the mechanism standing behind this apparent decreased electron output from PSII? Recently, it was shown that the excitation pressure, caused by an over-reduced PQ pool, results in over-reduction of PSII's $Q_A$ and $Q_B$. The increased formation of $Q_A^-$ changes the dissociation constant of the $HCO_3^-$ molecule, and can cause its release[46]. Later on, the unoccupied site favors an alternative attachment of an $O_2$ molecule[47]. At this point, a re-reduced semi-quinol at $Q_A^-$ cannot transfer the electron all the way to $Q_B$, since the non-heme iron molecule is attached to $O_2$. It is therefore probable that this $O_2$ molecule will undergo reduction by $Q_A^-$ and release the oxidative radical $O_2^{•-}$[47,48], which could later be converted to peroxide. This process will undoubtedly diminish the apparent electron output of PSII, as can be observed here (Figs. 1c, 2c and 4c). This observation coincides with the observed decreased activity of the purified complexes under anoxic conditions in the presence of an excess of oxidized quinone (DCBQ) (Fig. 3). However, we also observed an additional slowdown in PSII activity under anoxia, taking place in the presence of an excess of bicarbonate (Fig. 3). In addition, one should keep in mind that the in vivo observations (Fig. 4) show that turning off this slowdown and gaining back the original 3-fold faster rate of PSII, requires a minimal incubation of 15 min in dark anoxia— quite long for a direct binding of a ligand to its target. Therefore, it seems that although modulations in the non-heme iron affinities to bicarbonate decrease the rate of the apparent linear electron flow, they also trigger additional changes, which could have a prolonged effect.

One hypothesis involves a conformation change, in which the $HCO_3^-$ molecule is replaced by a glutamate residue of PsbD (E241-D2). Recently, such a structure was shown to exist in immature cyanobacterial PSII[49]. This was possible due to an attachment of another subunit, termed Psb28 (which share some functional similarities with the plant PsbW sub-unit[50]), that forms a strong bond to a loop on PsbA. Such distortion pushes the E241-D2 residue towards the non-heme iron and stabilizes the complex. It was also postulated that such a conformation might assist the assembly process of PSII by enabling residual electron transfer. In this regard it should be noted that the position of the E241-D2 next to the non-heme iron, is structurally similar to that of E234-M from the anoxygenic bacterial reaction center[51] (for further comparison, see refs. [41,52,53]). We could postulate then, that such changes are feasible under anoxia and might stand behind the altered activity rate. It could also explain why the relaxation of the complexes back to their optimal activity takes a long period of time. However, to date, such conformation was not detected in isolated PSII complexes, and so we could only speculate on its existence and possible function.

Alternatively, once the photosynthetic electron transport chain is over-reduced and a decoupling between PSII electron output

and $O_2$ evolution is established[54], One can suggest that the slowdown mechanism is not a direct radiative charge back-reaction, but rather an emergence of cyclic electron flow in PSII presumably via $Cytb_{559}$. Accordingly, the reduced heme at the $Cytb_{559}$ re-introduces the electrons to the chlorophyll pair of P680, towards the $Q_A$ site, in a manner which is dependent on the potential state of the heme[55]. Indeed, $Cytb_{559}$ was postulated to be able to oxidize plastoquinol from the $Q_B$ site and thus alleviate PSII acceptor limitations[56], and recent research correlated between decreased activity of the OEC with the augmentative electron flow via this cyclic pathway[57]. However, the mechanism of such a route remains elusive and will require further studies in the future. In any case, the added electron pressure will be alleviated by an alternative local electron acceptor, which could possibly be $O_2$ in the proximity of PSII.

In conclusion, we describe here an unexpected mechanism diminishing PSII electron output under anoxia. This process may prevent the overexcitation of the photosynthetic apparatus thereby mitigate oxidative damage. Revealing its mechanistic details will, of course, require additional experimental evidence and theoretical considerations. Yet, unfortunately, this decrease in the apparent linear electron flow also diminishes the electron output of the entire photosynthetic apparatus, posing new barriers to cope with in the search for photosynthesis-based renewable energy sources. As it stands now, this mechanism is the glass ceiling diminishing photosynthetic $H_2$ production yield to only a third of its potential.

## Methods

**Cells type, growth, and conditions.** *Chlamydomonas reinhardtii* strain CC124 (137c mt-) was obtained from the Chlamydomonas Resource Center. Cell cultures were maintained and cultivated in Erlenmeyer flasks under continuous illumination (at an irradiance of 60 μE m$^{-2}$ s$^{-1}$) in TAP medium (or TP for autotrophic cultures). Cell samples were taken from cultures in early log phase (between 2–5 mg Chl mL$^{-1}$). Chlorophyll concentration was determined using 90% acetone according to Ritchie[58].

**Cell preparation for photosynthetic measurements.** Cells were centrifuged to a final concentration of 20 mg Chl mL$^{-1}$ in TAP (or TP for autotrophic cultures), 50 mM HEPES, pH 7.2, and placed in a sealed quartz cuvette (5 mL). When indicated, glucose oxidase (200 units mL$^{-1}$), catalase (200 units mL$^{-1}$), and glucose (50 mM) were added to scavenge traces of $O_2$. Anaerobic induction was achieved by keeping the cells in darkness for an hour. If indicated, 40 μM (final concentration) of 3-(3,4-dichlorophenyl)-1,1-dimethylurea (DCMU, Sigma-Aldrich) and 1 mM (final concentration) of hydroxylamine (HA, Sigma-Aldrich) were added 10 min prior to the measurements. In all the experiments that were conducted in the JTS, 10% of Ficoll was added prior to induction, to avoid cell sedimentation.

**Combined gas exchange and photosynthetic efficiency measurements.** For tracking the concentration of diffused gasses, such as $H_2$ and $O_2$, the experimental cuvette was placed in a home-built membrane inlet mass spectrometer (MIMS), as described in ref. [16]. Simultaneously, Chl *a* fluorescence measurements were conducted using a DUAL-PAM-100 (Heinz Walz Gmbh, Effeltrich, Germany). Monitoring state transition was conducted by taking cell samples directly from the experimental cuvette (50 μL) and injecting them into a glass capillary and inserting it to a glass Dewar (Horiba) filled with liquid nitrogen (at 77 °K). Samples were then measured in a fluorimeter (Horiba Fluoromax-4), with excitation light was of 435 nm. Emission was measured between 650 and 750 nm.

**Spectroscopic analysis of the redox potential and electron flux.** Electrochromic shifts (ECS) were detected using a Joliot Type Spectrometer (JTS-100, Biologic SAS, France), supplied with a BiLED, which is able to simultaneously measure the absorbance of 520 and 546 nm, as described in ref. [59]. When indicated, laser flashes were pumped by a frequency-doubled Nd:YAG laser (Litron nano), excitation wavelength was adjusted using a dye (DCM, exciton laser dye). When cells are exposed to a single turnover laser flash their absorbance is changed via a 3-phase shift[27]. The initial increase in the ECS signal (termed "phase a") lasts less than a millisecond and is a product of charge separations taking place in both photosystems, hence decreased signal can report on their degradation. During the second phase (termed "phase b"), which usually lasts up to 10 ms[28], ECS signal increases due to the build-up of

membrane potential, mainly via $Cytb_6f$. In the final phase (termed "phase c"), the signal decreases exponentially due to the breakup dissipation of the membrane potential via ATPase activity. Since no differences in the amplitude of "phase a" were observed (Supplementary Figure 3), the results were normalized it. Observed differences in the kinetics of both the apparent "phase b" and "phase c" were used to determine shifts in the redox potential across the thylakoid membrane.

**Thermoluminescence measurements.** The cells were centrifuged shortly, transferred to a minimal medium (TP) in a final concentration of 15 mg Chl mL$^{-1}$. Anaerobic cultures were placed in a 13 mL hypovial bottles, and flushed with $N_2$ for 10 min under darkness, followed by an additional hour of dark incubation with continuous shaking. Illuminated cultures were subjected to 2 min of light exposure (at an irradiance of 370 μE m$^{-2}$ s$^{-1}$), followed by either 5 or 15 min of darkness before they were sampled (300 μL). TL measurements were carried out by a custom-made TL apparatus, as described in ref. [60]. For the measurements, the samples were placed on a copper plate in air, connected to a cold finger immersed in liquid $N_2$. A heater coil (SEI 10/50, Thermocoax, France) ensured the desired temperature of the sample during the measurement. It should be noted that in the case of whole cells, freezing before TL measurements is not recommended due to possible cellular damage[35], therefore algal samples were excited at 4 °C, by two single turnover saturating Xe flashes (of 1.5 μs duration at half-peak intensity, with a 1-s delay in-between flashes). Following this, the sample was heated to 70 °C in darkness with a heating rate of 20 °C min$^{-1}$, The emitted TL was measured with a photomultiplier (H10721-20, Hamamatsu, Japan) simultaneously with recording the temperature.

**PSII isolation.** *Chlamydomonas reinhardtii* cells were grown in a 10 L TAP medium. The cells were cultured with constant stirring and air bubbling under continuous white light (35–40 μE m$^{-2}$ s$^{-1}$) at 18 °C until final OD$_{730}$ reached 0.5–0.7. Then, the culture was checked for $O_2$ evolution, harvested by centrifugation at 3500 × $g$ for 5 min, and resuspended in a medium containing 25 mM HEPES pH 7.5, 300 mM sucrose and 5 mM MgCl$_2$. The cells were washed once in the same buffer, centrifuged at 5000 × $g$ for 5 min and resuspended in a main buffer containing 25 mM MES-NaOH, pH 6.5, 1 mM MgCl$_2$, 10 mM NaCl, 1 M betaine, 200 mM sucrose, and 10% of the glycerol. Protease-inhibitors cocktail was added to final concentrations of 1 mM PMSF, 1 μM pepstatin, 60 μM bestatin and 1 mM benzamidine. The cells were disrupted by an Avestin EmulsiFlex-C3 at 2000 psi (two cycles). Unbroken cells and starch granules were removed by centrifugation at 12,000 × $g$ for 5 min and the membranes in the supernatant were precipitated by centrifugation in a Ti70 rotor at 273,300 × $g$ for 40 min. The pellet was resuspended in the same breaking buffer giving a chlorophyll concentration of 2 mg Chl mL$^{-1}$. n-Decyl-α-D-Maltopyranoside (α-DM) and n-octyl β-D-glucopyranoside were added dropwise to a final concentration of 2.5% each and final chlorophyll concentration 1 mg Chl mL$^{-1}$. After stirring at 4 °C for 30 min, the insoluble material was removed by centrifugation at 12,000 × $g$ for 10 min. Supernatant was loaded on sucrose gradients in SW-60 rotor (≈900 μg of chlorophyll per tube), gradient composition 15–40% sucrose, 25 mM MES-NaOH, pH 6.0, 0.5 M betaine, 10% glycerol, 0.2% α-DM and run at 310,000 × $g$ for 14–16 h. The two lower bands were collected and measured directly (freezing is not possible as sample quality deteriorate rapidly) as described below.

**PSII activity determination.** For $O_2$ evolution determination, purified PSII at a final concentration of 10 μg chl mL$^{-1}$ was reconstituted in 25 mM MES-NaOH, pH 6.5, 2.5 mM MgCl$_2$, 2.5 mM CaCl$_2$, 1 M betaine, 12.5% Glycerol, 10 mM NaHCO$_3$, 0.05% α-DM supplemented with freshly made 350 μM 2,6-Dichloro-1,4-benzoquinone (DCBQ) to a final volume of 1 mL, and placed in an ALGi thermoregulated glass vial, capped with a silicon stopper. The $O_2$ concentration was obtained using an optical $O_2$ sensor (Pyroscience FireSting, OXROB3 probe). The reaction mixture was exposed to either: 5, 10, 20, or 30 s of illumination (370 μE m$^{-2}$ s$^{-1}$, white LED light) with a minute dark period in between exposures. In order to establish anaerobic conditions, a constant stream of an $N_2$ gas was injected into the vial's headspace through an inlet needle. The $O_2$ levels were monitored for a few minutes, until $O_2$ concentration was reduced to 10 μM. After which, the needle was ejected, and the experiment initiated as described above.

**Statistics and reproducibility.** For generating biological replications, cell cultures were inoculated from single colonies before each experiment. Since Figs. 1, 2 and 4 panels a–c describe a kinetic phenomena, the results are presented in their original traces, of the averaged results, to improve figure visibility and clarity. The repetitions showed the same ratio of change as described and presented in the text. Purified PSII complexes were harvested from isolated thylakoids prior to each experiment to prevent activity deterioration. Single repetitions and their averaged result are presented in the attached supplementary material file (Supplementary data 1). Significant differences in all Figures between initial and successive light exposures were

calculated using a student $t$ test, in Microsoft Excel, as well as the presented Box plots in Figs. 3 and 4d.

## Data availability

All the data are shown in this MS, Excel files of the raw data are available as a supplementary material (Supplementary data 1). In case of any further request, additional data can be shared with anyone who is interested.

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

## Acknowledgements
We thank M. Hippler and F. Buchert from WWU Münster, Germany for constructive discussion. We thank N. Shahar from TAU, Israel for carefully reading the manuscript and for his comments and László Kovács (BRC Szeged) for his assistance with the TL measurements. This work was supported by the Lendület/Momentum Programme of the Hungarian Academy of Sciences (LP2014/19 research grant to S.Z.T.) and the National Research, Development, and Innovation Office (K132600 and FK 135633) research grants to S.Z.T. and V.N.), and by the Israel Science Foundation (grant number 941/22 to I.Y.).

## Author contributions
Y.M., S.Z.T., and I.Y. designed research; Y.M, T.E, V.N., and I.Y. performed research; M.F. provided purified PSII; Y.M., S.Z.T., and I.Y. analyzed data; Y.M., S.Z.T., and I.Y. authored the paper.

## Competing interests
The authors declare no competing interests.
