## [Peer Review File · Communications Biology]

Reviewers' comments:

Reviewer #1 (Remarks to the Author):

Manuscript #: COMMSBIO-22-3045-T

A novel PSII photosynthetic control is activated in anoxic cultures of green algae

General Comment:

This manuscript describes a very urgent topic in the production of the photosynthetic hydrogen gas (H₂)—photosynthetic H₂ evolution, as a potential renewable source of biofuel. There exist many limitations that hamper the industrial scale production despite the extensive research in this area with the electrons' loss in the downstream metabolic processes as well as the susceptibility of the hydrogenase enzyme, which is responsible from the H₂ production to the oxygen. Extensive studies are currently underway to overcome these challenges with some degree of success as highlighted by the authors in the introduction. In the present manuscript the authors identified a limiting factor that was not known before which is an activation of a slow-down switch in photosystem II which occurs only under anoxic condition—a redox activated photosynthetic control, and showed that the photosynthetic control limits the H₂ production in the model *Chlamydomonas reinhardtii* grown under anoxic condition. In particular, the authors investigated the reason(s) behind the drastic drop on the electron fluxes by tracking the H₂ evolution process in both the intact algal cultures and the isolated photosystem II and concluded that the redox activated photosynthetic control causes a slowdown in the cyt b6f activity, which resulted in dramatic conformational changes at the acceptor side leading to a defect in the linear electron flow from the PSII down to the PSI and downstream processes. This is very intriguing and support the significance of the present work, however, the limitation which was identified by the authors is somewhat ambiguous and very difficult to extract from the current version of the manuscript. This and other concerns have to be addressed unambiguously before considering this manuscript for publication.

Major issues to be clarified:

- 1) In line 72: the authors have selected only one light exposure time (2 min) with somewhat high photon flux (370 $\mu\text{E}/\text{m}^2/\text{Sec}$) after an hour of dark anoxia incubation and no control data was presented. It is known *Chlamydomonas* strains vary in their responses to light intensity; did the authors estimated the response of the anoxic *C. reinhardtii* to photoinhibition? Perhaps a PAM Chla fluorescent measurement for a control group is needed here to exclude any possibility of photosynthetic decay due to photodamage. Please also note that the data presented in Figure 2a show that the electron output drops after longer light-exposure time >30 sec up to 3 min, which also show a negative effect to the longer light exposure after 1-hour dark anaerobiosis.
- 2) In Figure 1a as stated in line 80-82: The observed 3-fold decrease in the H₂ evolution rates in the successive light exposure in comparison with the initial exposure could be a result of photoinhibition to PSII which may lead to turn down in the electron flow, hence the H₂ evolution. If this isn't the case what was the control measurement to exclude this possibility?
- 3) Line 82-83: Can the authors highlight the time differences (triggering gap) of the O₂ evolution triggering of the initial and successive light exposure? Please clarify this point explicitly in the text.
- 4) In Figure 2 and Line 148: Was the 3 min dark incubation during the MIMS measurements also under aerobiosis or anoxia? This should be clarified.
- 5) One direct approach to limit the electron flow output would be by using an acceptor site inhibitor for example DCMU which binds competitively to the plastoquinone side and block the electron transfer and negatively affect the redox poise state. What was the control experiment during Chla fluorescence and the ECS experiments?
- 6) Line 189 and Figure 2e: it is not clear how the authors controlled the anoxic conditions for the isolated PSII complex? In any case and as mentioned in this line (see also comment 1) that the anoxic condition affects the PSII activity which necessarily cause defect in the electron flow output of PSII leading ultimately to the decrease in the H₂ production as the PSII is the main source of electrons for the hydrogenase enzyme. It is also important to show the O₂ evolution data of the control and the

dark anoxia PSII after 180 sec.

7) Line 234-238: what was the delay time between those two flashes (Δt)? it is important to have a delay time to allow for the reaction to occur before illuminating with the second flash at least $\Delta t = 0.25\text{ms}$. This need to be clarified.

8) In line 337-338: the authors stated that the redox pressure causes an intrinsic shift in PSII, which results in a decreased linear electron flow; while this is true but it doesn't exclude the possibility that such a shift in PSII may occur due to damage resulting from the long light exposure or the anoxic condition. Can the author elaborate more on this point?

Reviewer #2 (Remarks to the Author):

Summary.

This work reports a significant decrease in activity of photosystem II induced by a short (seconds) illumination of green algal cells when under anaerobic conditions. This does not occur in aerobic conditions. The effect is reversible upon dark adaptation for 15 minutes. The effect was discovered while studying H₂ generation (ref 7) but not located to PSII. In this work the main phenomena are well demonstrated using kinetic mass spectrometry to measure H₂ and O₂ production, extending the earlier work. In addition, fluorescence-based measurements of PSII activities were presented. A series of tests were also performed (fluorescence quenching states, state transitions, electrochromic changes), that monitor the main regulatory processes of the antenna and of photosynthetic electron transfer. The results of these tests are consistent with the activity drop occurring in PSII.

To verify this, PSII was then isolated and studied under comparable conditions, and it was found that a similar activity decrease.

Having localized the activity drop to PSII, the authors turned to the literature for a mechanistic explanation. They found one "ready cooked" in the 2016 report that bicarbonate is lost from its Fe binding site in PSII when QA⁻ is generated and when the bicarbonate concentration is low (43). This works as a protection and regulatory system in PSII (ref 43) and involves O₂ reduction at the vacant bicarbonate site on the non-heme iron (41). The authors also adopt an older model of cyclic electron transfer within PSII and combine the two mechanisms together.

Overall Impression.

The phenomena seem convincing and the findings are very interesting. The survey of potential explanations comes together to make a reasonable case that the usual regulatory mechanisms can be ruled out. The experiments with TL and those on isolated PSII argue in favor of a specific PSII effect. Some technical aspects can be raised (see below) but overall, the experiments (replicates etc) and arguments seem sound. The connection to the two literature models is reasonable and justified, although combining them is less so, and is confusing (see below). However, having put forward these potential explanations, specific experiments should be done that are aimed at demonstrating they do explain the current experimental regime. This would complete this very interesting study and significantly strengthen the work and increase its impact.

Below we make suggestions that should be quick and easy (on a good day).

Specific comments.

Major points

1) The absence of experiments testing for a role for bicarbonate is an obvious omission. It is known that other carboxylic acids can replace bicarbonate resulting in a significant slow-down of electron transfer (e.g. [https://doi.org/10.1016/0005-2728\(94\)90045-0](https://doi.org/10.1016/0005-2728(94)90045-0)) and the effects of bicarb loss are reversed by re-addition bicarbonate. Experiments can be done using these effects.

a) Note: these effects occur in vivo (<https://doi.org/10.1104/pp.18.00341> and <https://doi.org/10.1016/j.bbabi.2013.06.004>). In Chlamy, when grown in mixotrophic conditions, acetate in the media appears to bind in the bicarb site. The inhibition reported here may be enhanced

by the presence of acetate competing with bicarbonate. We note that the inhibition is significantly smaller in the autotrophic cultures in supp figure 1, this could be an indication that bicarb release is involved, but specific experiments are called for.

b) In ref 43, the bicarbonate release occurs not only when QA- is oxidized but also when the bicarbonate concentration is low. To make a firm connection to that mechanism, the mass spec system may be used to verify any changes in the CO₂ concentration as a proxy for the bicarbonate.

2) The article also lacks experiments (and few arguments) that favor the cyclic pathway occurring under the conditions of the present study.

3) The blending together of both the bicarbonate-release mechanism and the cyclic mechanism is unjustified and confusing

4) The different methods used to achieve anaerobicity could have different side effects: degassing will remove both O₂ and CO₂ (and thus bicarbonate, depending on the pH). This could influence some of the experiments if bicarbonate is involved. This could be tested.

5) The TL experiment is not very convincing. The effect is clear but not easily interpretable. The background slope is problematic. The dark incubated TL (dotted light gray according to the legend) is invisible. Control experiments in the presence of DCMU should identify the position of the peak associated with the S₂/3QA- state. Control experiments using ionophores and uncouplers could in principle allow you to assess any contribution of the $\Delta\psi$ and ΔpH on the peaks position in the different experimental conditions.

Other and minor points:

1) Title: statements as titles should be discouraged. It is often non-scientific to make blunt statements without the appropriate conditionals. Just remove the "is". Think about it.

2) Abstract: put in the biological material used.

3) Line 16 and 17. Unsupported statements of faith and hype should be avoided. The first sentence should be toned down and made more accurate. I suggest "Photosynthetic hydrogen production from microalgae is considered to have potential as a renewable energy source". This statement is closer to what you say in the text, is true and it also avoids sounding like a direct sales pitch.

4) Line 21: change "slashing" to "reducing" (same reasoning as other point 2).

5) Line 25-26: this point is not made in the text. Cut it from the abstract or discuss it in the text in sufficient detail to make the suggestion reasonable.

6) Line 48 and 49: cut "highly effective" as it is not that. Put in "potential" as it may qualify as that.

7) Alternations means "the repeated occurrence of two things in turn". It is used several times in the text. This meaning is not evident. Alterations, seems more appropriate.

8) Line 107: correct the spelling of "conformationnel" to "conformational".

9) Line 301: what is the evidence for a conformational change? Clarify, argue better or be less specific about a conformational change.

10) Line 309: ref 43 discovered this, ref 40 was an attempt to confirm it.

11) Line 321: Cut "was later proved that indeed". It is inaccurate. The citation is to a (later) review, in which the much earlier the experiments indicating electrons from QBH₂ can reach b559 are cited.

12) The figure is very "schematic", perhaps "inaccurate" describes it better. But the bigger problem is that it is not really justified by the findings. It is a rather dubious hybrid of the various mechanisms taken from the literature. If the tests for a role for bicarbonate confirm its role, then the figure (preferable a more accurate one) would become more justified.

13) A detailed read through should allow the correction unwieldy constructions, and the removal of surplus words.

Reviewers A W Rutherford and A Fantuzzi

Reviewer #4 (Remarks to the Author):

The authors aim at understanding the origin of the decrease of linear electron flow in photosynthesis in fluctuating light after anoxia conditions. The decrease of electrons flux in photosynthesis in

successive exposures to light limits H₂ production. They assess the electron flow in intact *Chlamydomonas* cells and in purified PSII complex using different spectroscopic techniques and mass spectrometry. The authors propose that the activation of a redox "photosynthetic control" at PSII alters the electron flow and consequently H₂ production. The presented results are interesting to understand photosynthesis in fluctuating light associated with anoxia conditions.

1. The authors suggest that under anoxia conditions, PSII regulates electron flow through the cyclic electron flow in PSII presumably via the Cytb559. To provide more robustness to the Cytb559 involvement in play, more experimental support is needed. One of the proposed functions for Cytb559 is indeed to keep the acceptor side of PSII oxidized, however other hypotheses have been made (ref 46 for example). But, Cytb559 function still remains elusive. It would be interesting to check if any changes in the redox potential of the Cytb559 can be detected by EPR spectroscopy. This experiment can be done on PSII and thylakoids, but maybe on cells as well. Another approach would be the study of H₂ production in *Chlamydomonas* mutants of the Cytb559. Some of the heme ligation mutants are viable (Morais et al., 2001, *J Biol Chem* 276, 31986-93; Hamilton et al., 2014 *Plant Cell Physiol* 55, 1276-85).
2. Cell cultivation in autotrophic conditions will make photosynthetic H₂ production sustainable. Despite higher rate of H₂ production in TAP medium, would not be more interesting to look at autotrophic growing cells? Are the observations made on PSII in TAP growing cells reproducible in TP medium or minimal medium?
3. Line 271: referring to Figure 3d (TL on cells) is written: "Error bars indicate standard error (n ≥ 3)". There are no error bars in TL glow curves. Otherwise, statistical data analysis is appropriate.
4. How do authors intend to reverse engineer the "PSII slow-down switch" to allow efficient photosynthetic production of H₂?

Minor comments

Line 20: "we" not "We".

Line 110: "successive" not "Successive".

Lines 195-196 (Figure 2): the text in the brackets needs rephrasing.

Line 199 (Figure 2): "were plotted against the accumulated concentration of H₂".

Line 207: "1st and 2nd exposures" and not "1st and 2nd exposures".

I suggest to check the paragraph of materials and methods. Some spaces are missing and some misspellings are found within.

Some mistakes in the use of Saxon genitive were found in the text.

A novel PSII photosynthetic control is activated in anoxic cultures of green algae

Your manuscript entitled "A novel PSII photosynthetic control is activated in anoxic cultures of green algae" has now been seen by 3 referees. You will see from their comments below that while they find your work of considerable interest, some important points are raised. We are interested in the possibility of publishing your study in *Communications Biology*, but would like to consider your response to these concerns in the form of a revised manuscript before we make a final decision on publication.

We therefore invite you to revise and resubmit your manuscript, taking into account the points raised. In particular, we ask that you add data from additional, control experiments requested by Reviewers #1 and #2, as well as perform experiments that directly test if bicarbonate (Reviewer #2) and Cytb559 (Reviewer #3) contribute to the observed inhibition of PSII activity. The reviewers also suggested a number of improvements to the text, which you should consider incorporating into your revised manuscript.

Reviewers comments, Manuscript #: COMMSBIO-22-3045-T

Reviewer #1:

General Comment:

This manuscript describes a very urgent topic in the production of the photosynthetic hydrogen gas (H_2) - photosynthetic H_2 evolution, as a potential renewable source of biofuel. There exist many limitations that hamper the industrial scale production despite the extensive research in this area with the electrons' loss in the downstream metabolic processes as well as the susceptibility of the hydrogenase enzyme, which is responsible from the H_2 production to the oxygen. Extensive studies are currently underway to overcome these challenges with some degree of success as highlighted by the authors in the introduction. In the present manuscript the authors identified a limiting factor that was not known before which is an activation of a slow-down switch in photosystem II which occurs only under anoxic condition—a redox activated photosynthetic control, and showed that the photosynthetic control limits the H_2 production in the model *Chlamydomonas reinhardtii* grown under anoxic condition. In particular, the authors investigated the reason(s) behind the drastic drop on the electron fluxes by tracking the H_2 evolution process in both the intact algal cultures and the isolated photosystem II and concluded that the redox activated photosynthetic control causes a slowdown in the *Cytb_{6f}* activity, which resulted in dramatic conformational changes at the acceptor side leading to a defect in the linear electron flow from the PSII down to the PSI and downstream processes. This is very intriguing and support the significance of the present work, however, the limitation which was identified by the authors is somewhat ambiguous and very difficult to extract from the current version of the manuscript. This and other concerns have to be addressed unambiguously before considering this manuscript for publication.

Major issues to be clarified:

1) In line 72: the authors have selected only one light exposure time (2 min) with somewhat high photon flux ($370 \mu\text{E m}^{-2} \text{sec}^{-1}$) after an hour of dark anoxia incubation and no control data was presented. It is known *Chlamydomonas* strains vary in their responses to light intensity; did the authors estimated the response of the anoxic *C. reinhardtii* to photoinhibition? Perhaps a PAM Chla fluorescent measurement for a control group is needed here to exclude any possibility of photosynthetic decay due to photodamage. Please also note that the data presented in Figure 2a show that the electron output drops after longer light-exposure time >30 sec up to 3 min, which also show a negative effect to the longer light exposure after 1-hour dark anaerobiosis.

- We thank the reviewer for his remarks. We will address the photosynthetic decay, in the answer to point number 2, as it deals with this question specifically. Concerning the other points, we elaborate here, that as this work is a continuation work for 3 published papers (Milrad et al., 2018; Kanygin et al., 2020; Milrad et al., 2021). The protocol which we devised for these measurements was originally carried out in various conditions and specifically light regimes prior to the set-up in the manuscript. Indeed, we observed that H_2 evolution is almost entirely **independent** to irradiance levels, in a w.t strain (see (Kanygin et al., 2020)). However, in contrast, O_2 evolution rate is. Therefore, we chose an irradiance at which the total O_2 accumulation during two minutes of exposures, isn't damaging or inhibiting hydrogenase's activity, do not cause photosynthetic damage, but still; triggers photosynthetic control. Therefore, the selected irradiance was chosen to be in-between, so some O_2 levels would be apparent, and convincing for the broad audience, while at the same time not too high to start inflicting photodamage.

2) In Figure 1a as stated in line 80-82: The observed 3-fold decrease in the H_2 evolution rates in the successive light exposure in comparison with the initial exposure could be a result of photoinhibition to PSII which may lead to turn down in the electron flow, hence the H_2 evolution. If this isn't the case what was the control measurement to exclude this possibility?

- We thank the reviewer for this very important comment. We have indeed at first considered the possibility of PSII photodamage, which would surely decrease electron flux. Yet our tests along the manuscript clearly show that this is not the case as:
 - a) As the reviewer suggested, using PAM we show (in revised Supp.6), the F_m values before (1 min) and after (30 sec) each light exposure. These results show no difference between aligned measurements, meaning that the regaining of the original values is fast. In contrast, regeneration of PSII activity following photodamage takes a long time, in the range of hours. It should be noted that under anoxia, cells do not get to the optimal 80% value, but normally set around ~40-50%. We have however added to the revised MS an explanation that will make this point clearer: "We could therefore conclude that the potential activity of PSII is not decreased due to photodamage or state transitions." (Lines: 245-247)

- b) To verify the integrity of the photosynthetic apparatus, we also used ECS measurements which act as a voltmeter and measure the total electron output from PSII and PSI. The measurements were carried out post laser flashes before each illumination (**Figure. 1d**), which show no difference in trends or values for the a-phase in the μsec range. This fast phase should show a marked decrease if the core PSII systems were damaged.
- c) Lastly, if the PSII cores faced damage during the initial exposure, then, the second and third exposures would also show a gradual decrease in all parameters, such as: H_2 and O_2 evolution, fluorescence etc. However, this is not the case and we have also previously demonstrated that no further changes can be observed even following 10 of such fluctuations (see Milrad et al. 2018).
- d) It is worth mentioning that we did referred to these explanations in the discussion: "PSII centers damage was shown as implausible by the fact that both the saturating pulses and the ECS measurements showed no differences between light fluctuations" (Lines: 354-355).

3) Line 82-83: Can the authors highlight the time differences (triggering gap) of the O_2 evolution triggering of the initial and successive light exposure? Please clarify this point explicitly in the text.

- We are grateful for this question, as can be observed here in the results, there is no triggering time, and O_2 starts evolving as soon as the light is turned on. We have discussed these issues in (Milrad et al., 2018). The difference between the observed O_2 accumulation kinetics remains obscure though and can be attributed to other O_2 consuming pathways. We mentioned it in the revised MS: "Interestingly, we also observed that the initial exposure triggered an immediate linear increase in net O_2 evolution, in contrast to successive exposures in which an exponential increase can be seen (**Figure. 1b**- dashed vs. solid)", (Lines: 83-85). We have also noted that this difference is not observed under autotrophic conditions (using TP as a medium, **Supp. 1**), which might hint on a different and not necessarily correlated metabolism. This observation was also made recently in (Fitzpatrick et al., 2022), where they show that O_2 evolution is triggered at light onset in a stable manner, and that its consumption rate decrease timely.

4) In Figure 2 and Line 148: Was the 3 min dark incubation during the MIMS measurements also under aerobiosis or anoxia? This should be clarified.

- We are sorry for this misunderstanding, in this experiment, we changed the duration of illumination, without changing the dark exposures in-between. Therefore, O_2 levels were roughly the same as in **Figure. 1** (considering that the amount of accumulated O_2 depends on the duration of illumination). It is important to note that in every test, O_2 levels drop rapidly as soon as the light is turned off (as can be seen in **Figure. 1b**). Therefore, in all tests, O_2 levels reached 0 before the successive fluctuations were implemented, and at no point of the experiment surpassed $6 \mu\text{M}$ of O_2 , which is way

below the threshold of hydrogenase inactivation (see also **Supp. 1**, where relative higher O₂ levels show the same phenotype, and **Figure. 3**, where the addition of O₂ scavengers do not alter it).

5) One direct approach to limit the electron flow output would be by using an acceptor site inhibitor for example DCMU which binds competitively to the plastoquinone side and block the electron transfer and negatively affect the redox poise state. What was the control experiment during Chla fluorescence and the ECS experiments?

- The effects of DCMU on the evolution of H₂ were thoroughly discussed in (Milrad et al., 2021), where we demonstrated the manner in which H₂ evolution is blocked in it's presence. The same observations were made by (Cournac et al., 2002), where they showed that DCMU interfere with H₂ evolution. We therefore did not include these results in this MS. In fact, the electron flux was shown to be severely decreased (as was measured by ECS in (Milrad et al., 2021)), and the evolution rate of H₂ is so small, that any differences or the lack of them, could not be statistically separated.

6) Line 189 and Figure 2e: it is not clear how the authors controlled the anoxic conditions for the isolated PSII complex? In any case and as mentioned in this line (see also comment 1) that the anoxic condition affects the PSII activity which necessarily cause defect in the electron flow output of PSII leading ultimately to the decrease in the H₂ production as the PSII is the main source of electrons for the hydrogenase enzyme. It is also important to show the O₂ evolution data of the control and the dark anoxia PSII after 180 sec.

- We are sorry for this misunderstanding however, since we did not want to go into technical details in the result section, we mentioned it in Material and Methods, revised MS: "The O₂ levels were monitored for a few minutes, until O₂ concentration was reduced to 10 μM, after which, the needle was ejected, and the experiment initiated as described above." (Lines: 505-506) We hope that this clarifies the concerns of the reviewer.
- It should be elaborated that purified PSII has a very short lifetime. For example, it is so sensitive that one cannot freeze and thaw it. In short, we had to work with freshly made samples immediately after collecting them from the sucrose gradient. Due to its sensitivity, minutes long experiments with purified PSII are not feasible as the sample is losing its activity which render long (minutes range) conclusions impossible. In addition, O₂ was removed from the medium using N₂ (for the anoxic treated enzymes) sparging, which was only implemented prior to each measurement, and not during or after illuminations. Therefore, by prolonging the illumination time we also increased O₂ levels which were accumulated during light exposures. Hence, in order to not reach aerobic levels, and maintain low O₂ levels for the successive illuminations, we were not able to further examine longer periods of light exposure.

7) Line 234-238: what was the delay time between those two flashes (Δt)? it is important to have a delay time to allow for the reaction to occur before illuminating with the second flash at least $\Delta t = 0.25\text{ms}$. This need to be clarified.

- We thank the reviewer for this comment, the delay time in-between flashes was of 1 second, therefore sufficient to ease these concerns. In order to clarify this point, we added to the methods section the explanation: “with a one second delay in-between flashes” (Lines: 469-470).

8) In line 337-338: the authors stated that the redox pressure causes an intrinsic shift in PSII, which results in a decreased linear electron flow; while this is true but it doesn't exclude the possibility that such a shift in PSII may occur due to damage resulting from the long light exposure or the anoxic condition. Can the author elaborate more on this point?

- Indeed, an excellent comment. As we explained here, we did not observe any indication of an actual PSII damage, contrary to our own initial estimations. Therefore, we started investigating other factor which might influence the efficiency of PSII. As mentioned in this work, we tested state transitions, using both PAM fluorescence and 77°K approaches, PSII photoinhibition (see comment 2) and many others before we finally came to the conclusions that formed this hypothesis. It is important to note that we raise a hypothesis which will need to be furtherly clarified in the future, hopefully by other study groups that can examine it using different approaches. In general, the effects of anoxic conditions on the activity of PSII were not sufficiently studied and several issues remain unattended, e.g. why do we see a decreased efficiency under anoxia? However, these questions cannot be resolved in the scope of this work. We do have some educated guesses, but it is a subject that deserves its own focus and we hope to elaborate on it in future papers.

Reviewer #2

Summary.

This work reports a significant decrease in activity of photosystem II induced by a short (seconds) illumination of green algal cells when under anaerobic conditions. This does not occur in aerobic conditions. The effect is reversible upon dark adaptation for 15 minutes. The effect was discovered while studying H₂ generation (ref 7) but not located to PSII. In this work the main phenomena are well demonstrated using kinetic mass spectrometry to measure H₂ and O₂ production, extending the earlier work. In addition, fluorescence-based measurements of PSII activities were presented. A series of tests were also performed (fluorescence quenching states, state transitions, electrochromic changes), that monitor the main regulatory processes of the antenna and of photosynthetic electron transfer. The results of these tests are consistent with the activity drop occurring in PSII. To verify this, PSII was then isolated and studied under comparable conditions, and it was found that a similar activity decrease.

Having localized the activity drop to PSII, the authors turned to the literature for a mechanistic explanation. They found one “ready cooked” in the 2016 report that bicarbonate is lost from its Fe binding site in PSII when Q_A⁻ is generated and when the bicarbonate concentration is low (43). This works as a protection and regulatory system in PSII (ref 43) and involves O₂ reduction at the vacant bicarbonate site on the

non-heme iron (41). The authors also adopt an older model of cyclic electron transfer within PSII and combine the two mechanisms together.

Overall Impression.

The phenomena seem convincing and the findings are very interesting. The survey of potential explanations comes together to make a reasonable case that the usual regulatory mechanisms can be ruled out. The experiments with TL and those on isolated PSII argue in favor of a specific PSII effect. Some technical aspects can be raised (see below) but overall, the experiments (replicates etc) and arguments seem sound. The connection to the two literature models is reasonable and justified, although combining them is less so, and is confusing (see below). However, having put forward these potential explanations, specific experiments should be done that are aimed at demonstrating they do explain the current experimental regime. This would complete this very interesting study and significantly strengthen the work and increase its impact.

Major points

1) The absence of experiments testing for a role for bicarbonate is an obvious omission. It is known that other carboxylic acids can replace bicarbonate resulting in a significant slow-down of electron transfer (Hamilton et al., 2014) and the effects of bicarb loss are reversed by re-addition bicarbonate. Experiments can be done using these effects.

a) Note: these effects occur in vivo (Roach et al., 2013; Messant et al., 2018). In *Chlamy*, when grown in mixotrophic conditions, acetate in the media appears to bind in the bicarb site. The inhibition reported here may be enhanced by the presence of acetate competing with bicarbonate. We note that the inhibition is significantly smaller in the autotrophic cultures in supp figure 1, this could be an indication that bicarb release is involved, but specific experiments are called for.

- We thank for this suggestion, as the reviewer suggested we indeed compared the effects of adding acetate. However, following a careful comparison between the mixotrophic and autotrophic conditions, we did not observe any significant differences. Our data show that the decrease of the accumulated H₂ concentration under autotrophic conditions is at $54 \pm 5\%$. However, since this can vary between samples we added to the figure (**Supp. 1**, revised MS) another measurement in the presence of O₂ scavengers (Gox). These measurements show a higher decrease in accumulated H₂ concentration, of $64 \pm 6\%$, which resembles that of the presented data for a mixotrophic culture. We hope that this added data will ease these concerns.

b) In ref 43, the bicarbonate release occurs not only when Q_A⁻ is oxidized but also when the bicarbonate concentration is low. To make a firm connection to that mechanism, the mass spec system may be used to verify any changes in the CO₂ concentration as a proxy for the bicarbonate.

- We thank for this comment, we have indeed measured changes in CO₂ concentrations during these light fluctuations, and presented these results in (Milrad et al., 2018).

There, we observed a sharp increase in CO₂ at light onset and attributed it to the activation of the CCM mechanism. Unfortunately, the scale of this increase is too large and probably masks the inner effects of PSII CO₂ release. However, we have observed also, that there is some difference between this CO₂ increase between the initial and successive exposures (see also (Milrad et al., 2018)). But, when we analyzed the data, and plotted the rate of CO₂ exchange in time, we saw that the rate in which CO₂ concentration decreases (due to its fixation by the CBB cycle) does not change between fluctuations. We could not however, determine the source of this difference and eliminate the option of altered CCM activation nor mitochondrial respiration (which could also explain the difference in O₂ accumulation rate). Finally, we decided to omit these results, as they only complicate the discussion and are generally hard to explain without going into excessive details.

2) The article also lacks experiments (and few arguments) that favor the cyclic pathway occurring under the conditions of the present study.

- We thank the reviewer for this good suggestion – following it, we studied the possible contribution of cyclic electron flow by adding the substance DMBQ, which previously was shown to inhibit PSII cyclic electron flow *in vivo* (Ananyev et al., 2017). We observed that although in the presence of DMBQ, H₂ evolution rates were decreased, further decline of H₂ production is eliminated. We also tested the cells under the same conditions using thermoluminescence and saw that following light exposure, the B-band value, did not regain its initial temperature even after long dark re-adaptation periods. We added these results in the revised MS (**Figure. 4**), and discuss their interpretation towards our proposed mechanism (Lines: 292-318)

3) The blending together of both the bicarbonate-release mechanism and the cyclic mechanism is unjustified and confusing.

- In this work, we observed that there is a decrease in PSII activity, under conditions where the PQ pool is known to be reduced. This state was postulated to trigger the release of HCO₃⁻ from the iron site in PSII and cause limitations on its acceptor side. It was indeed evidences also, that in the absence of CO₂ in the medium, isolated PSII complexes do not face such limitations, regardless to the depletion of O₂ from the medium (**Supp. 5**). Therefore, we could postulate that the mechanism could involve the suggested release of HCO₃⁻, in accordance to previous work of others (Shevela et al., 2020; Fantuzzi et al., 2022). It was indeed also postulated that upon such release, an O₂ molecule could substitute HCO₃⁻ and be further reduced by Q_A⁻ which will result in a release of electron from the over reduced PSII complex. However, the source of the electron which function in that reaction was not attributed, as evidenced in these publications, and was also postulated to be Cytb₅₅₉, in accordance with the hypothesis made by (Bondarava et al., 2010).
- Our results, in the presence of DMBQ, point on a formation of CEF within PSII. And since Qc was not observed to be present in algal or plant PSII, the only proposed

mechanism that we can rely on to date involves *Cytb₅₅₉*. Therefore, taking into account that we obtained evidence to both a CO₂ dependent inactivation and CEF formation, under our anaerobic conditions, and that both processes seems to be activated and retrieved concomitantly, we feel comfortable enough to propose our suggested mechanism. It should be noted that we do propose this mechanism as a hypothesis in our set conditions, and that this should not necessarily reflect the situation under different conditions, in which only one part or another might arise, e.g. under light adapted anoxia or conditions which might trigger different states of *Cytb₅₅₉*. Yet in the scope of our observations, we strongly feel that with the available research so far, the proposed mechanism cannot be overruled.

4) The different methods used to achieve anaerobicity could have different side effects: degassing will remove both O₂ and CO₂ (and thus bicarbonate, depending on the pH). This could influence some of the experiments if bicarbonate is involved. This could be tested.

- We are grateful for this comment, as the reviewer suggested, we performed this control (complete omission of bicarbonate) only with purified PSII (see revised MS **Supp. 5**). In addition, we added to the results section an explanation for it: “To assess the role of CO₂ (which was added in the form of NaHCO₃, see material and methods), we conducted the same experiment in its complete absence. In line with the literature, we observed a slower rate of O₂ production (65% of the rate in the presence of CO₂)³². However, we have not seen a difference between light fluctuations, be it under aerobic or anoxic conditions (**Supplementary Fig. 5**). These results hint on the role of CO₂ or HCO₃⁻ in the observed diminished rate of electron output from PSI.”. (Lines: 192-198). It should be elaborated that the complete omission of bicarbonate is possible only with purified PSII and not *in vivo* as cellular conditions will always have some bicarbonate. To our surprise the anoxic slow down phenomenon in purified PSII, was not observed in the complete absence of bicarbonate. Although it is highly intriguing, we suggest that this is an outcome of a synthetic non-natural situation.

5) The TL experiment is not very convincing. The effect is clear but not easily interpretable. The background slope is problematic. The dark incubated TL (dotted light gray according to the legend) is invisible. Control experiments in the presence of DCMU should identify the position of the peak associated with the S₂/3QA- state. Control experiments using ionophores and uncouplers could in principle allow you to assess any contribution of the Δψ and ΔpH on the peaks position in the different experimental conditions.

- We thank the referees for their comments. As the experiments were carried out on intact cells and we observed the B band. Upon DCMU treatment, the B band is eliminated, and the so-called Q band appears with a peak position at around 0 °C. For properly measuring it, the samples need to be frozen. However, upon freezing, the TL intensity strongly diminishes and the peak positions may also change (Janda et al.,

2004). Therefore, the suggested experiment is unfortunately not suitable to identify the position of the peak associated with the $S_{2/3} Q_A^-$ state.

- Concerning the dark aerobic samples, we thank the reviewer for noticing that. Indeed, the graph is not visible, since we eliminated it from the manuscript, in order to simplify it. These results do not add any conclusions to the work, and therefore were deemed unnecessary. Unfortunately, we missed the line in which it is stated in our proofing and removed it in the revised version.

Other and minor points:

1) Title: statements as titles should be discouraged. It is often non-scientific to make blunt statements without the appropriate conditionals. Just remove the “is”. Think about it.

- Done.

2) Abstract: put in the biological material used.

- Line 23 revised MS: “techniques on *Chlamydomonas reinhardtii* cultures, “

3) Line 16 and 17. Unsupported statements of faith and hype should be avoided. The first sentence should be toned down and made more accurate. I suggest “Photosynthetic hydrogen production from microalgae is considered to have potential as a renewable energy source”. This statement is closer to what you say in the text, is true and it also avoids sounding like a direct sales pitch.

- Done.

4) Line 21: change “slashing” to “reducing” (same reasoning as other point 2).

- Rephrased to limiting

5) Line 25-26: this point is not made in the text. Cut it from the abstract or discuss it in the text in sufficient detail to make the suggestion reasonable.

- rephrased

6) Line 48 and 49: cut “highly effective” as it is not that. Put in “potential” as it may qualify as that.

- rephrased

7) Alternations means “the repeated occurrence of two things in turn”. It is used several times in the text. This meaning is not evident. Alterations, seems more appropriate.

- rephrased

8) Line 107: correct the spelling of “conformationnel” to “conformational”.

- rephrased

9) Line 301: what is the evidence for a conformational change? Clarify, argue better or be less specific about a conformational change.

- rephrased

10) Line 309: ref 43 discovered this, ref 40 was an attempt to confirm it.

- Changed

11) Line 321: Cut “was later proved that indeed”. It is inaccurate. The citation is to a (later) review, in which the much earlier the experiments indicating electrons from QBH2 can reach b559 are cited.

- rephrased

12) The figure is very “schematic”, perhaps “inaccurate” describes it better. But the bigger problem is that it is not really justified by the findings. It is a rather dubious hybrid of the various mechanisms taken from the literature. If the tests for a role for bicarbonate confirm its role, then the figure (preferable a more accurate one) would become more justified.

- We thank the reviewer for noticing some inaccuracies concerning the exact localization of the relevant subunits in our presented scheme. Since we would like to draw interest to the manuscript, we modified the figure so now, in the new version, we used the PDB structure of dimer PSII (nr. 6KAF). We hope that this will satisfy the concerns made by the reviewer

13) A detailed read through should allow the correction unwieldy constructions, and the removal of surplus words.

- We thank the reviewers for their concerns and will follow their instructions.

Reviewers A W Rutherford and A Fantuzzi

Reviewer #4

Summary.

The authors aim at understanding the origin of the decrease of linear electron flow in photosynthesis in fluctuating light after anoxia conditions. The decrease of electrons flux in photosynthesis in successive exposures to light limits H₂ production. They asses the electron flow in intact Chlamydomonas cells and in purified PSII complex using different spectroscopic techniques and mass spectrometry. The authors propose that the activation of a redox “photosynthetic control” at PSII alters the electron flow and consequently H₂ production. The presented results are interesting to understand photosynthesis in fluctuating light associate with anoxia conditions.

Major points

1) The authors suggest that under anoxia conditions, PSII regulates electron flow through the cyclic electron flow in PSII presumably via the Cytb559. To provide more robustness to the Cytb559 involvement into play, more experimental support is needed. One of the proposed function for Cytb559 is indeed to keep the acceptor side of PSII oxidized, however others hypothesis have been made (ref 46 for example). But, Cytb559 function still remains elusive. It would interesting to check if any changes

in the redox potential of the Cytb559 can be detected by EPR spectroscopy. This experiment can be done on PSII and thylakoids, but maybe on cells as well. Another approach would be the study of H₂ production in Chlamydomonas mutants of the Cytb559. Some of the heme ligation mutants are viable (Morais et al., 2001, J Biol Chem 276, 31986-93; Hamilton et al., 2014 Plant Cell Physiol 55, 1276-85).

- We thank the reviewer for his suggestion. Sadly, following a correspondence with Peter Nixon who generated the strain, it appears that this mutant was lost and cannot be stored for long time. Furthermore, as the mutant he was refereeing to accumulates no more than 20% of active PSII, the results will be incomparable to the w.t strain. Therefore, we have chosen an alternative method to address his concerns. Since Cytb₅₅₉ is hypothesized to be involved in PSII cyclic electron flow, we studied it by adding the substance DMBQ, which previously was shown to inhibit PSII cyclic electron flow *in vivo* (Ananyev et al., 2017). We observed that in the presence of DMBQ, the slow down effect is minimized either by measuring H₂ production or by measuring Chl fluorescence. We added these results to the revised MS (**Figure. 4**).

2) Cell cultivation in autotrophic conditions will make photosynthetic H₂ production sustainable. Despite higher rate of H₂ production in TAP medium, would not be more interesting to look at autotrophic growing cells? Are the observations made on PSII in TAP growing cells reproducible in TP medium or minimal medium?

- We agree with the referee, indeed it is very interesting to test these schemes under autotrophic conditions. However, it does add some technical problems, especially in considerations of CO₂ assimilations. Cells which are grown under minimal medium tends to grow slow if they are exposed to only ambient CO₂ from the surrounding air. Therefore, in many cases, we tend to bubble the cultures with a CO₂ mix (5%). Yet, since CO₂ levels were opted to play an important role in this study, we were concerned that these fine tunes of growth conditions might perplex the readers and generate objections. We have in fact, conducted many studies using minimal medium to grow the cells. In these studies, the rate of H₂ evolution, O₂ accumulation and CO₂ fixations are naturally different than those we observe, when cells are grown and exposed to acetate. However, the described change due to the exposure to light and the decrease that we observe in successive illuminations remains (see **Supp. 1**). Therefore, harshening the conditions for the cells does not grant us additional conclusions, and might complicate the scheme's reproducibility by other research groups. Taking all of the above into account, we resolved to present in this work our main findings, that assist us in explaining the reasoning of our proposed mechanism.

3) Line 271: referring to Figure 3d (TL on cells) is written: "Error bars indicate standard error (n ≥ 3)". There are no error bars in TL glow curves. Otherwise, statistical data analysis is appropriate.

- We thank the reviewer for noticing that error, in a former version of the figure we thought of presenting these results in a numerical way. But after several

considerations, we decided that a graphical presentation would be more appealing to the audience. Unfortunately, we left the legend as it is. In the revised MS, we added another supplemental figure (**Supp. 7**), that shows the results for all 5 flashes that we conducted on the cells under these conditions. There, we show the results with the addition of error bars for each point. In addition, the new added **Figure.4**, that show the effect of DMBQ addition also show box plots, in which the data distribution is presented.

4) How do the authors intend to reverse engineering the “PSII slow-down switch” to allow efficient photosynthetic production of H₂?

- We share the referee’s interest – we have some future plans that hopefully will be fruitful, in a way that will grant us the ability to manipulate algae’s metabolism, and by doing so, resolving one of the major holdbacks of bio-H₂ production. But naturally this MS describes a phenomenon – not a solution. We do hope that our findings here will encourage other researchers to address these issues, and of course are always open for further discussions on that subject.

Minor comments

Line 20: “we” not “We”.

- rephrased

Line 110: “successive” not “Successive”.

- rephrased

Lines 195-196 (Figure 2): the text in the brackets needs a rephrasing.

- rephrased

Line 199 (Figure 2): “were plotted against the accumulated concentration of H₂”.

- rephrased

Line 207: “1st and 2nd exposures” and not “1st and 2nd exposures”.

- rephrased

I suggest to check the paragraph of materials and methods. Some spaces are missing and some misspelling are found within. Some mistakes in the use of Saxon genitive were found in the text.

- We thank the reviewer for the comment and we went through the text again.

Reviewers' comments:

Reviewer #1 (Remarks to the Author):

The authors have adequately addressed all of the major concerns raised in my previous report and I believe that the revised manuscript has been well improved.

Reviewer #2 (Remarks to the Author):

Reviewer #2

Summary:

The authors have addressed most of the question raised. There is good news and bad news. First the bad news. As it stands and as written, the lack of experimental evidence associated with HCO₃⁻ and the Cytb559 cyclic pathway, the model given here remains unjustified and misleading. The model should be removed. However, it is perfectly acceptable to make a statement saying that the phenomenon reported here may be related to the literature model in which bicarb is lost when QA⁻ is present (Brinkert et al 2016), or more speculatively, to some inefficiency due to the Cyt b559 pathway, may be mentioned. There is no problem linking observations to the literature, but the authors should refrain from unjustified speculation.

Now the good news. In Point 3, the existing data may be re-interpreted, and with one or two minor experiments of the same kind already presented, the association with bicarbonate may be demonstrated. These data seem likely to confirm the bicarb role (or they could eliminate it). fits with a quite specific, experimentally demonstrated, and well understood mechanism (Brinkert et al 2016). I urge the authors to take the chance and get it over the line. It is worth a try.

Below we answer the points made in detail hoping to inform the authors and to give them a better feel for the subject and literature as well as making suggestions to improve the present paper.

Specific replies:

Point 1 We suggested experiments with bicarbonate (or other carboxylic acids) to directly test the bicarb model. In the answer no direct experiments on bicarbonate were done (but see points 3 and 4).

Point 1a We also pointed out that acetate and glycolate can affect PSII electron transfer in vivo. This could explain the difference in the H₂ decrease in mixotrophic vs auto trophic growth conditions as shown in the article. Instead of doing experiments, the authors replied saying there is no difference between autotrophic growth and mixotrophic growth. That may be the case, but they do not show it. Instead, they added another autotrophic experiment but with O₂ scavengers. This may be okay, although it is not convincing. A more convincing response would be a statistical treatment of the variation in the data for H₂ decreases in mixotrophic vs autotrophic growth. An explanation for the very different magnitudes of H₂ concentrations between mixo and auto trophic conditions would be useful too.

Point 1b. First, we apologise for our writing error: we said "when QA⁻ is oxidized" rather than "when QA⁻ is present". We hope (and assume) there was little confusion, given the clarity of literature on this subject.

In answer to the suggestion that the mass spec could be used to measure HCO₃⁻ by monitoring CO₂ in solution, the authors state it is not easy. That we can fully accept. We also accept that changing bicarbonate concentrations in the cells may be complicated by the complexity of the living system. But it might have been worth a try. And, given the effect of acetate has been reported in Chlamy, then that would have been a good place to start.

Point 2 We and reviewer 4 stated that there was no evidence in the article in favour of cyclic electron transfer in PSII.

The additional experiments purported to answer this question focused on using DMBQ, a less common,

but well-used electron acceptor from PSII. The experiments demonstrate clearly that when PSII can give its electrons to an exogenous acceptor, then it overcomes electron flow bottlenecks. It is not surprising that this can result in improved forward e⁻ transfer through PSII. However, it provides no evidence for cyclic electron transfer. If cyclic electron transfer were taking place, (and there is no evidence that it does), the addition of an electron acceptor would deprive it of electrons too. This would also be manifest as improved forward electron transfer. But that would not be evidence for the existence of cyclic e⁻ transfer in PSII. All of these phenomena can be explained simply by the alleviation of the forward electron transfer restriction (arising from overreduction of the PQ due to downstream bottle necks) under these anoxic condition.

We suggest the authors keep the experiment but change the interpretation: drop the cyclic idea. The citation given, on which the approach and interpretation (cyclic) is based, should be reassessed in this context.

Comment on new TL data with DMBQ.

It has been known for decades that the addition of PSII electron acceptors (or mediators, or oxidants, or reductants) remove or diminish TL due to removal of the reduced (or oxidized) species (in this case QA⁻ or QB⁻) preventing radiative recombination. The nature of the effect depends on the experimental conditions, with, for example, incomplete effects seen when there are kinetic limitations (in whole cells for example). It is doubtful that the TL presented here provide any information other than the trivial and well known fact that removal of electrons by an exogenous acceptor diminishes the chances of radiative decay of charge pairs in PSII. We suggest this data and its discussion is removed. It weakens the study.

The small fraction of TL remaining somehow escapes the added electron acceptor. What this sub population represents has not been tested. The most likely explanation is that the PMF has been collapsed by the quinone (hence the higher temperature). But it could be a minor sub-population that is in a strange environment in the cell. In any case there appears to be no evidence here for a relationship to the putative cyclic electron transfer.

In conclusion there is still no experimental reason to suggest a role of cyclic e⁻ transfer in PSII in this work. This is a weakness.

Point 3: The mixture of the bicarb model and cyclic is unjustified and confusing.

The answer to this point is unclear.

- The authors first point:

The logic of experiments in Supp Fig 5 (O₂ evolution on the second illumination in isolated PSII without bicarbonate addition, +/- O₂ depletion,) is poorly explained (see point 4). It was very hard to decipher what has been done based on the information provided. Here correction have been made to help in comprehension. In Supp 5, O₂ removal (by gassing with N₂) is shown to have little effect on O₂ evolution after a pre-illumination. It is stated " It was indeed shown, that in the absence of HCO₃⁻ in the medium, isolated PSII complexes do not face such limitations,....." (that is, the slower e⁻ transfer after the first illumination). This seems important.

Crucially, it is mentioned in the text and quoted in point 4 (but not in point 3) that: "we observed a slower rate of O₂ production (65% of the rate in the presence of HCO₃⁻)." when bicarbonate was omitted from the isolated prep. This activity drop seems to be due to bicarbonate depletion when bicarb is omitted. Despite these 2 important observations, no clear conclusions are made although they make vague connections to a role of bicarbonate.

In fact, in point 4, the authors remain confused over what is being observed, as demonstrated by their statement: "To our surprise the anoxic slow down phenomenon in purified PSII, was not observed in the complete absence of bicarbonate." They go on to laugh it off as an artefact.

And yet the absence of an inhibition effect on the first illumination, would be exactly what would be expected if the bicarbonate loss (due to QA⁻ formation, as in Brinkert et al 2016) were responsible for the inhibitory effect reported here.

The key experiment needed to verify this result would be the re-addition of bicarbonate to these bicarbonate-depleted samples. This should lead to the recovery of the original activity and the light-induced loss of activity on the first illumination. To see this, bicarbonate would not only need to be re-added, but also subsequently the concentration of bicarbonate in the buffer would need to be lowered so the concentration of bicarb falls to a value that is between the dissociation constants estimated for the QA and QA⁻ states (Brinkert et al 2016). Under these conditions the light-induced formation of QA⁻ should result in bicarbonate release and the consequent decrease in O₂ evolution.

Given the light-induced inhibition was seen in isolated PSII in the presence of bicarbonate, then it must be assumed that the N₂ gassing procedure in this buffer (the pH) and the timing is sufficient to lower the bicarb concentration into the zone where QA⁻ formation results in bicarbonate dissociation.

In the answer to the point 3, the authors are imprecise and confused when it comes to the literature.

i) The authors state that their postulate is "in accordance with Shevela et al 2020 and Fantuzzi et al 2022".

In fact, they should have referenced the key work on this (Brinkert et al 2016), in which QA⁻ induced release of bicarbonate was discovered and the mechanism proposed. The Shevela et al paper was a follow-up attempting to verify bicarbonate release via mass spec of CO₂.

ii) The authors go on to state "It was indeed also postulated that upon such release, an O₂ molecule could substitute HCO₃⁻ and be further reduced by QA⁻ which will result in a release of electron from the over-reduced PSII complex. However, the source of the electron which function in that reaction was not attributed, as evidenced in these publications, and was also postulated to be Cytb559, in accordance with the hypothesis made by (Bondarava et al., 2010)."

This first sentence is poorly written, but the second sentence is a misreading of the papers in question. In the Fantuzzi et al 2022 paper, to which they refer, the QA⁻ was monitored directly and was shown to undergo oxidation when O₂ was added, with a rate dependent on the O₂ concentration. This reaction was blocked by bicarbonate binding. The amount of superoxide formed by this reaction was stoichiometric with the QA⁻ oxidized. This paper clearly does assign the origin of the electron, its product and it also argues that the effect of bicarbonate implicates the non-heme iron, vacated by the bicarbonate, as the binding site for the O₂ where it undergoes reduction. This is then supported by computational chemistry. The sources, origin and of the electron are probably better attributed here than in any previous study on this subject.

The confusion here presumably arises from the additional (side) observation in Fantuzzi et al 2022 that some superoxide is formed that does not correlate with QA⁻ oxidation. The donor in this case is not identified (and was not directly relevant to the main theme of that paper, as it was not the electron coming from QA⁻) and the usual suspects were mooted, i.e., contaminating PSI, Pheo-, Cytb559, and PQH₂ pool.

In passing, Bondarava et al, as mentioned by the authors, reported low potential Cyt b 559 being capable of oxidizing PQH₂ and generating O₂^{•-}. However, there was no discussion of cyclic electron flow in Bondara et al.

- The authors second comment answering point 3: the effect of DMBQ.

See comments in point 2 above. The DMBQ data do not seem to provide data relevant to cyclic e⁻ flow through PSII.

Point 4: removal of O₂ and inadvertent removal of CO₂ and thus bicarbonate.

This is partially dealt with in point 3. Although the specific point is not really tested except in the data of Supp 5, with isolated PSII omitting bicarbonate from the suspension medium.

To make Figure Supp 5 more comprehensible as figure and the paper more useful as a contribution, several changes should be made.

- i) Correct bicarb formula in the legend
- ii) Correct the title from "low carbon" to: in the absence of bicarbonate).
- iii) Call it bicarbonate rather than CO₂ or carbon, as this is the relevant species involved in Fe binding (and it is much less confusing).
- iv) The figure should contain the comparable data with i) the samples containing bicarb as the control with 100% activity (plus light induced inhibition), ii) without bicarb with 65% activity (no light induced inhibition) and iii) bicarb added back to ii) showing recovery of 100% activity and light-induced inhibition. Or just ii) and iii)
- v) Assuming, re-addition bicarb results in recovery of maximum activity and the light-induced activity decrease (under the right experimental conditions), this figure would be the most important of the paper. Thus, it should be in the main text.

Point 5

We expressed concern about the TL study.

The arguments given in answer are weak and unconvincing. They do not make a useful contribution to the paper.

Minor Points:

Nearly all minor points were answered appropriately however the figure 5 is still problematic.

The figure somewhat improved. However, given there is no evidence for cyclic electron flow, then the figure as drawn is misleading. Assuming the authors agree with the interpretation of Sus Fig 5 given in point 3 above, then the Brinkert et al 2016 model would be appropriate: i) over-reduction the PQ pool, results in ii) over-reduction of the QA and QB (forming QA- QBH₂ formation), iii) the presence of QA- changes the dissociation constant of the bicarbonate, iv) releasing it when its concentration in the medium is low enough. The model of Fantuzzi et al 2022, comes into play, with v) O₂ binding the empty site on the non-heme iron, vi) where it accepts the electron from QA- and vii) is released as superoxide.

The rest of the diagram is not needed. However, for future reference, charge separation is usually considered to take place from ChlD1 (read reference 43)

Step 2 as shown and described very oversimplified and mistaken. The electron donation from carotenoidD2 likely comes in via ChlD2 (read Faller,P., Fufezan C and Rutherford AW (2005) in Photosystem II:The Light Driven Water:Plastoquinone Oxidoreductase pp. 347-365 (T Wydrzynski and K. Satoh eds) Springer). This reference will give the authors more information for understanding the cyclic electron transfer in PSII. Note, the donation from the TyrZ and the Mn cluster is not out competed under any known physiological conditions (see Faller et al 2005).

Step 3 in the diagram and as described in legend is also misleading. It does not correspond the literature models, although it could occur on subsequent steps after those listed above, in some conditions. Here it is confusing, probably wrong, and should not be included.

A W Rutherford and A. Fantuzzi

Reviewer #4 (Remarks to the Author):

In the re-submitted manuscript, Milrad and coworkers present new results to support their hypothesis. The authors propose that PSII reduces the electron flow in photosynthesis and consequently H₂ production in *Chlamydomonas* in fluctuating light after anoxia conditions. H₂ is considered as a potential renewable energy source.

General comment

Newly added experiments confirm that the decrease of H₂ production in the studied conditions depends on a drop in PSII activity. However, the mechanism that cause the decrease of PSII activity is not clear. Authors favors the idea of cyclic electron flow within PSII via Cytb559 that takes electrons from the reduced plastoquinone pool. Between the new experiments, authors study the effect of DMBQ on Chlamydomonas cells. The addition of DMBQ oxidize the acceptor site of photosystem II thereby reducing back reactions in PSII and cyclic electron flow within PSII. The authors claim there is no back reactions but indeed, it would be useful to perform thermoluminescence measurement in presence of DCMU to measure them. I wonder if DMBQ is just removing cyclic electron flow within PSII or generally alleviating reduction of plastoquinone pool. Generally, I still think there are not supported experimental proofs that cyclic electron flow within PSII via Cytb559 is operating in these conditions.

Regarding the reduction of O₂ by QA⁻ that forms O₂•⁻ and later hydrogen peroxide, it will interesting to measure them.

Minor comments

Line 216-217. Sentence was just partially corrected.

Line 416 "Cells; type, growth, and conditions ". The ";" is not needed.

I suggest to indicate that NaHCO₃ is used both in fig 2e and in the text. In this way, readers will avoid to check in material and methods.

Line 362. The ";" is not needed.

Supplemented Figure. 5. NaHCO₃ not NaCO₃

Point to point replies

Reviewer #1

The authors have adequately addressed all of the major concerns raised in my previous report and I believe that the revised manuscript has been well improved.

Reviewer #2

Summary:

The authors have addressed most of the question raised. There is good news and bad news. First the bad news. As it stands and as written, the lack of experimental evidence associated with HCO_3^- and the *Cytb₅₅₉* cyclic pathway, the model given here remains unjustified and misleading. The model should be removed. However, it is perfectly acceptable to make a statement saying that the phenomenon reported here may be related to the literature model in which bicarb is lost when Q_A^- is present (Brinkert et al 2016), or more speculatively, to some inefficiency due to the *Cytb₅₅₉* pathway, may be mentioned. There is no problem linking observations to the literature, but the authors should refrain from unjustified speculation.

- We accept the reviewer's suggestion and modified the presented mechanism accordingly. In our revised discussion, we describe three different mechanisms which can explain together or separately the slowdown mechanism observed under anoxia in the presence of an excess of bicarbonate.

Now the good news. In Point 3, the existing data may be re-interpreted, and with one or two minor experiments of the same kind already presented, the association with bicarbonate may be demonstrated. These data seem likely to confirm the bicarb role (or they could eliminate it). fits with a quite specific, experimentally demonstrated, and well understood mechanism (Brinkert et al 2016). I urge the authors to take the chance and get it over the line. It is worth a try. Below we answer the points made in detail hoping to inform the authors and to give them a better feel for the subject and literature as well as making suggestions to improve the present paper.

- We are thankful for the extremely constructive review and appreciate the time invested by the reviewer to improve our MS. Please see our specific responses below.

Specific replies:

Point 1

We suggested experiments with bicarbonate (or other carboxylic acids) to directly test the bicarb model. In the answer no direct experiments on bicarbonate were done (but see points 3 and 4).

- We apologize for the misunderstanding, as our answers were not to the point of this comment. In the revised MS we examined the effects of bicarbonate, in resemblance to what was suggested (see revised Figure. 3).

Point 1a

We also pointed out that acetate and glycolate can affect PSII electron transfer in vivo. This could explain the difference in the H₂ decrease in mixotrophic vs autotrophic growth conditions as shown in the article. Instead of doing experiments, the authors replied saying there is no difference between autotrophic growth and mixotrophic growth. That may be the case, but they do not show it. Instead, they added another autotrophic experiment but with O₂ scavengers. This may be okay, although it is not convincing. A more convincing response would be a statistical treatment of the variation in the data for H₂ decreases in mixotrophic vs autotrophic growth. An explanation for the very different magnitudes of H₂ concentrations between mixo and auto trophic conditions would be useful too.

- We thank the reviewer for this constructive comment, per his request, we added a new figure (Sup. 1e) and text showing the differences.:
“Notably, the H₂ production rates were lower under such conditions, as we previously reported (Milrad et al., 2018). Since such conditions stimulate higher rates of O₂ evolution, The accumulated O₂ increase the competitive inhibition of hydrogenase activity (by reactions such as ‘Mehler-like’, Mehler and others (Burlacot et al., 2018)). Therefore, we conducted these measurements in the presence or the absence of O₂ scavengers. Interestingly, the addition of such scavengers slightly increased the accumulated H₂, from 10.7 ± 1.7 μM H₂ in their absence to 14.4 ± 1.3 μM H₂ in their presence (roughly a half of the value accumulated under mixotrophic adapted cells, 32.6 ± 2.5 μM H₂). In addition, we observed a stark decline in H₂ production rates, between the first and successive light exposures (for both treatments), which stood at 50 ± 1% and 65 ± 6% inhibition, in the absence or presence of O₂ scavengers, “

Point 1b

First, we apologize for our writing error: we said “when Q_A^- is oxidized” rather than “when Q_A^- is present”. We hope (and assume) there was little confusion, given the clarity of literature on this subject. In answer to the suggestion that the mass spec could be used to measure HCO_3^- by monitoring CO_2 in solution, the authors state it is not easy. That we can fully accept. We also accept that changing bicarbonate concentrations in the cells may be complicated by the complexity of the living system. But it might have been worth a try. And, given the effect of acetate has been reported in Chlamy, then that would have been a good place to start.

- We thank this suggestion, but as we replied in the previous revision, we believe that it not technically possible for very low concentrations in our MIMS (quadrupole) system to detect such complex changes as these changes are within the edge of detection and highly sensitive to minor numerous factors taking place simultaneously. We do however add MIMS data in reply to point 3. There we show how a drastic acidic treatment (adding HCl in an excess) can be used to evaluate the actual concentration of bicarbonate after N_2 sparging. Using this method converting all the bicarbonate to CO_2 by a strong acid (Supp Fig. 5) we estimated the bicarbonate concentration following the N_2 sparging that we used in the purified PSII experiments showing that it was 7.5mM (from a starting concentration of 10mM).

Point 2

We and reviewer 4 stated that there was no evidence in the article in favor of cyclic electron transfer in PSII. The additional experiments purported to answer this question focused on using DMBQ, a less common, but well-used electron acceptor from PSII. The experiments demonstrate clearly that when PSII can give its electrons to an exogenous acceptor, then it overcomes electron flow bottlenecks. It is not surprising that this can result in improved forward e^- transfer through PSII. However, it provides no evidence for cyclic electron transfer. If cyclic electron transfer were taking place, (and there is no evidence that it does), the addition of an electron acceptor would deprive it of electrons too. This would also be manifest as improved forward electron transfer. But that would not be evidence for the existence of cyclic e^- transfer in PSII. All of these phenomena can be explained simply by the alleviation of the forward electron transfer restriction (arising from overreduction of the PQ due to downstream bottle necks) under these anoxic condition.

- We accept this suggestion, removed the DMBQ data. However, as the reviewer will see in the reply to point 3 below, due to lack of evidence to confirm or reject the involvement of cyclic e^- flow, we still suggest it as an alternative out of three presented in our revised discussion.

We suggest the authors keep the experiment but change the interpretation: drop the cyclic idea. The citation given, on which the approach and interpretation (cyclic) is based, should be reassessed in this context

- We thank for this remark however, due to the strong objection of this reviewer as presented below, we removed the DMBQ data entirely.

Comment on new TL data with DMBQ.

It has been known for decades that the addition of PSII electron acceptors (or mediators, or oxidants, or reductants) remove or diminish TL due to removal of the reduced (or oxidized) species (in this case Q_A^- or Q_B^-) preventing radiative recombination. The nature of the effect depends on the experimental conditions, with, for example, incomplete effects seen when there are kinetic limitations (in whole cells for example). It is doubtful that the TL presented here provide any information other than the trivial and well known fact that removal of electrons by an exogenous acceptor diminishes the chances of radiative decay of charge pairs in PSII. We suggest this data and its discussion is removed. It weakens the study.

The small fraction of TL remaining somehow escapes the added electron acceptor. What this sub population represents has not been tested. The most likely explanation is that the PMF has been collapsed by the quinone (hence the higher temperature). But it could be a minor sub-population that is in a strange environment in the cell. In any case there appears to be no evidence here for a relationship to the putative cyclic electron transfer.

In conclusion there is still no experimental reason to suggest a role of cyclic e^- transfer in PSII in this work. This is a weakness.

- We accept these remarks, as suggested, all of the DMBQ data was removed and its related discussion.

Point 3

The mixture of the bicarb model and cyclic is unjustified and confusing. The answer to this point is unclear. The authors first point: The logic of experiments in Supp Fig 5 (O_2 evolution

on the second illumination in isolated PSII without bicarbonate addition, +/- O₂ depletion,) is poorly explained (see point 4). It was very hard to decipher what has been done based on the information provided. Here correction have been made to help in comprehension. In Supp 5, O₂ removal (by gassing with N₂) is shown to have little effect on O₂ evolution after a pre-illumination. It is stated “ It was indeed shown, that in the absence of HCO₃⁻ in the medium, isolated PSII complexes do not face such limitations,.....” (that is, the slower e⁻ transfer after the first illumination). This seems important.

- We are sorry for this misunderstanding, a new figure 3a is added (in the updated version, panel e. of figure 2 was modified to panel b. of figure 3). As the reviewer will see, the data suggests an anoxic slowdown switch taking place in PSII. However, it cannot be used to conclude what the fine details of the mechanism within PSII are. Furthermore, we strongly feel that it is far beyond the scope of this work. The new figure (**Figure 3**) shows the kinetics of PSII in the presence or absence of 10mM (actual 7.5mM see Sup. Fig 5) bicarbonate following 10 seconds of illumination. We chose this duration as it is the minimal duration of irradiance that activates the slowdown mechanism. While the results show no difference w/wo bicarbonate under aerobic conditions, they do show:

- 1) Dark Anoxic slowdown PSII seen on the first illumination w/wo bicarbonate
- 2) An additional slowdown during the second illumination taking place only in the presence of bicarbonate.

To explain “1”; We and other have seen this phenomenon *in vivo*; there, it was previously explained by the over reduction of the plastoquinone pool under anoxia. However, this explanation is not valid here, as we have plenty of oxidized quinone (DCBQ). We don't have a clear understanding why. However, it does hints that something else happens to PSII under dark anoxia. While, we think that it is an extremely interesting phenomenon, we feel it is beyond the scope of this work which describes a light dependent anoxic slowdown. As to “2”, we see an additional slow down only in the presence of 7.5mM bicarbonate. In the discussion, we suggest three possible mechanisms which can act separately or together to initiate the observed light dependent anoxic slowdown. In general, our conclusion is that PSII behavior and regulation under anoxia is a vast unexplored ocean that need to be thoroughly studied in the years to come.

Crucially, it is mentioned in the text and quoted in point 4 (but not in point 3) that: “we observed a slower rate of O₂ production (65% of the rate in the presence of HCO₃⁻.)” when bicarbonate was omitted from the isolated prep. This activity drop seems to be due to bicarbonate depletion when bicarb is omitted. Despite these 2 important observations, no clear conclusions are made although they make vague connections to a role of bicarbonate.

- We are sorry for this confusion, as described above we added new figure (**Figure. 3**) and further assessed the data. No differences w/wo bicarbonate are observed under aerobic conditions. The only difference (further slowdown) is seen only under anoxia, in the presence of bicarbonate.

In fact, in point 4, the authors remain confused over what is being observed, as demonstrated by their statement: “To our surprise the anoxic slow down phenomenon in purified PSII, was not observed in the complete absence of bicarbonate.” They go on to laugh it off as an artefact. And yet the absence of an inhibition effect on the first illumination, would be exactly what would be expected if the bicarbonate loss (due to Q_A^- formation, as in Brinkert et al 2016) were responsible for the inhibitory effect reported here. The key experiment needed to verify this result would be the re-addition of bicarbonate to these bicarbonate-depleted samples. This should lead to the recovery of the original activity and the light-induced loss of activity on the first illumination. To see this, bicarbonate would not only need to be re-added, but also subsequently the concentration of bicarbonate in the buffer would need to be lowered so the concentration of bicarb falls to a value that is between the dissociation constants estimated for the Q_A and Q_A^- states (Brinkert et al 2016). Under these conditions the light-induced formation of Q_A^- should result in bicarbonate release and the consequent decrease in O_2 evolution. Given the light-induced inhibition was seen in isolated PSII in the presence of bicarbonate, then it must be assumed that the N_2 gassing procedure in this buffer (the pH) and the timing is sufficient to lower the bicarb concentration into the zone where Q_A^- formation results in bicarbonate dissociation.

- We thank the reviewer for these remarks. To remedy the reviewer concerns, we added a MIMS data (Supp Fig. 5) showing that following the N_2 sparging and flushing all of the CO_2 , the bicarbonate concentration was at 7.5mM, 3 orders of magnitude higher than the μM range of the dissociation constant. Therefore, the suggested re-addition experiment would not lead to any new findings. As explained above, anoxia triggers two defined slowdowns; the first, taking place in the dark, is seen at the first light instance either in the presence or absence of 7.5mM bicarbonate. The second is seen during the second light instance, only in the presence of bicarbonate. In the revised discussion we provided three optional mechanisms to explain this phenomenon.

In the answer to the point 3, the authors are imprecise and confused when it comes to the literature.

i) The authors state that their postulate is “in accordance with Shevela et al 2020 and Fantuzzi et al 2022”. In fact, they should have referenced the key work on this (Brinkert et al 2016), in which Q_A^- induced release of bicarbonate was discovered and the mechanism proposed. The Shevela et al paper was a follow-up attempting to verify bicarbonate release via mass spec of CO_2 .

- We thank this remark and we corrected the text accordingly.

ii) The authors go on to state “It was indeed also postulated that upon such release, an O_2 molecule could substitute HCO_3^- and be further reduced by Q_A^- which will result in a release of electron from the over-reduced PSII complex. However, the source of the electron which function in that reaction was not attributed, as evidenced in these publications, and was also postulated to be *Cytb₅₅₉*, in accordance with the hypothesis made by (Bondarava et al., 2010).”

This first sentence is poorly written, but the second sentence is a misreading of the papers in question. In the Fantuzzi et al 2022 paper, to which they refer, the Q_A^- was monitored directly and was shown to undergo oxidation when O_2 was added, with a rate dependent on the O_2 concentration. This reaction was blocked by bicarbonate binding. The amount of superoxide formed by this reaction was stoichiometric with the Q_A^- oxidized. This paper clearly does assign the origin of the electron, its product and it also argues that the effect of bicarbonate implicates the non-heme iron, vacated by the bicarbonate, as the binding site for the O_2 where it undergoes reduction. This is then supported by computational chemistry. The sources, and origin of the electron are probably better attributed here than in any previous study on this subject. The confusion here presumably arises from the additional (side) observation in Fantuzzi et al 2022 that some superoxide is formed that does not correlate with Q_A^- oxidation. The donor in this case is not identified (and was not directly relevant to the main theme of that paper, as it was not the electron coming from Q_A^-) and the usual suspects were mooted, i.e., contaminating PSI, Pheo⁻, *Cytb₅₅₉*, and PQH₂ pool.

- We are thankful for the detailed explanation; the discussion was modified accordingly.

In passing, Bondarava et al, as mentioned by the authors, reported low potential *Cytb₅₅₉* being capable of oxidizing PQH₂ and generating $O_2^{\cdot-}$. However, there was no discussion of cyclic electron flow in Bondarava et al. The authors second comment answering point 3: the effect

of DMBQ. See comments in point 2 above. The DMBQ data do not seem to provide data relevant to cyclic e- flow through PSII.

- We agree with the reviewer's judgment and removed the DMBQ data and related discussion.

Point 4

Removal of O₂ and inadvertent removal of CO₂ and thus bicarbonate.

This is partially dealt with in point 3. Although the specific point is not really tested except in the data of Supp 5, with isolated PSII omitting bicarbonate from the suspension medium.

To make Figure Supp 5 more comprehensible as figure and the paper more useful as a contribution, several changes should be made.

i) Correct bicarb formula in the legend

ii) Correct the title from "low carbon" to: in the absence of bicarbonate).

iii) Call it bicarbonate rather than CO₂ or carbon, as this is the relevant species involved in Fe binding (and it is much less confusing).

iv) The figure should contain the comparable data with i) the samples containing bicarb as the control with 100% activity (plus light induced inhibition), ii) without bicarb with 65% activity (no light induced inhibition) and iii) bicarb added back to ii) showing recovery of 100% activity and light-induced inhibition. Or just ii) and iii)

v) Assuming, re-addition bicarb results in recovery of maximum activity and the light-induced activity decrease (under the right experimental conditions), this figure would be the most important of the paper. Thus, it should be in the main text.

- We are thankful for the detailed suggestions. Since the figure has a more important role to the paper, we divided the experiments to two panels, now presented as a new figure (**Figure. 3**). We also addressed his concerns regarding the concentration of bicarbonate in (**Supplement Figure. 5**).

Point 5

We expressed concern about the TL study. The arguments given in answer are weak and unconvincing. They do not make a useful contribution to the paper.

- We respect the reviewer comments and accept almost all of them. Having said this, we don't agree on this one, the TL data in the absence of DMBQ shows nicely that the time needed to regain the fast electron transfer from PSII is 15 minutes in the dark. In our view, it pinpoints *in vivo* that the source of the anoxic slowdown to PSII. This is in line with the same observation made with the other methods. Therefore, we decided to keep it.

Minor Points

Nearly all minor points were answered appropriately however the figure 5 is still problematic.

The figure somewhat improved. However, given there is no evidence for cyclic electron flow, then the figure as drawn is misleading. Assuming the authors agree with the interpretation of Sup Fig 5 given in point 3 above, then the Brinkert et al 2016 model would be appropriate:

- i) over-reduction the PQ pool, results in
- ii) over-reduction of the Q_A and Q_B (forming Q_A^- Q_BH_2 formation),
- iii) the presence of Q_A^- changes the dissociation constant of the bicarbonate,
- iv) releasing it when its concentration in the medium is low enough. The model of Fantuzzi et al 2022, comes into play, with
- v) O_2 binding the empty site on the non-heme iron,
- vi) where it accepts the electron from Q_A^- and
- vii) is released as superoxide.

The rest of the diagram is not needed. However, for future reference, charge separation is usually considered to take place from ChlD1 (read reference 43)

Step 2 as shown and described very oversimplified and mistaken. The electron donation from carotenoidD2 likely comes in via ChlD2 (read Faller,P., Fufezan C and Rutherford AW (2005) in Photosystem II:The Light Driven Water:Plastoquinone Oxidoreductase pp. 347-365 (T Wydrzynski and K. Satoh eds) Springer). This reference will give the authors more information for understanding the cyclic electron transfer in PSII. Note, the donation from the TyrZ and the Mn cluster is not out competed under any known physiological conditions (see Faller et al 2005).

Step 3 in the diagram and as described in legend is also misleading. It does not correspond the literature models, although it could occur on subsequent steps after those listed above, in some conditions. Here it is confusing, probably wrong, and should not be included.

A W Rutherford and A. Fantuzzi

- We are more than grateful for the productive discussion, as it seems, the mechanism of this inhibition is not as strikingly clear as we wished it to be, and therefore, we removed the figure (previously Figure. 5) and instead modified the discussion so it would include three options, as the reviewer suggested.

Reviewer #4

Summary

In the re-submitted manuscript, Milrad and coworkers present new results to support their hypothesis. The authors propose that PSII reduces the electron flow in photosynthesis and consequently H₂ production in *Chlamydomonas* in fluctuating light after anoxia conditions. H₂ is considered as a potential renewable energy source.

General comment

Newly added experiments confirm that the decrease of H₂ production in the studied conditions depends on a drop in PSII activity. However, the mechanism that cause the decrease of PSII activity is not clear. Authors favors the idea of cyclic electron flow within PSII via *Cytb₅₅₉* that takes electrons from the reduced plastoquinone pool. Between the new experiments, authors study the effect of DMBQ on *Chlamydomonas* cells. The addition of DMBQ oxidize the acceptor site of photosystem II thereby reducing back reactions in PSII and cyclic electron flow within PSII. The authors claim there is no back reactions but indeed, it would be useful to perform thermoluminescence measurement in presence of DCMU to measure them. I wonder if DMBQ is just removing cyclic electron flow within PSII or generally alleviating reduction of plastoquinone pool. Generally, I still think there are not supported experimental proofs that cyclic electron flow within PSII via *Cytb₅₅₉* is operating in these conditions. Regarding the reduction of O₂ by Q_A⁻ that forms O₂^{•-} and later hydrogen peroxide, it will be interesting to measure them.

- We thank the reviewer for the productive work, as both reviewers expressed concerns regarding our model, we modified it. Now the modified discussion includes three options, as the reviewers suggested.

Minor comments

Line 216-217. Sentence was just partially corrected.

- Modified in the revision

Line 416 “Cells; type, growth, and conditions “. The “;” is not needed.

- corrected

I suggest to indicate that NaHCO_3 is used both in fig 2e and in the text. In this way, readers will avoid to check in material and methods.

- Added to Figure 3 legend: “The complexes were tested in the absence or presence of 10mM (actual 7.5mM) NaHCO_3 to mimic high carbon conditions, ...”
- Line 215: “In addition, to assess the effects of bicarbonate (which was added in the form of NaHCO_3 , see materials and methods), we conducted the same experiment in its presence or complete absence.”

Line 362. The “;” is not needed.

- corrected

Supplemented Figure. 5. NaHCO_3 not NaCO_3

- corrected

REVIEWERS' COMMENTS:

Reviewer #2 (Remarks to the Author):

The authors have addressed our comments in great detail, with admirably open-minds, and much patience. Their balanced responses, additional experiments and positivity has resulted in major improvements in the article.

The authors chose to maintain the TL data against our suggestions (point 5): we fully agree with the authors decision. Our suggestion to remove all the TL was due to our (blinkered) view of the TL data as only being related to the cyclic electron flow hypothesis. Its utility in the present context is well justified.

We are happy to recommend publication.

A W Rutherford and A Fantuzzi

Reviewer #4 (Remarks to the Author):

In this manuscript, the authors aim at understanding the origin of the decrease of linear electron flow in photosynthesis in fluctuating light after anoxia conditions. The decrease of electrons flux in photosynthesis after successive exposures to light limits H₂ production, a potential renewable energy source. Experiments presented in the manuscript suggest that the decrease of H₂ production in the studied conditions depends on a drop in PSII activity. The mechanism that cause the decrease of PSII activity is investigated.

In the revised manuscript, Milrad and coworkers address some of the questions previously raised.

General comment

New added experiments show that bicarbonate affects the PSII activity, reducing the electrons coming out from PSII. This is not specific of anoxic condition but it is an interesting result.

Authors suggest that an over-reduction of the PQ-pool increases the formation of QA^{•-} that has the effect to displace the bicarbonate from the non-heme iron of PSII. The unoccupied site at the non-heme iron is taken by O₂, which in turn receives an electron by QA^{•-} forming O₂^{•-}. This mechanism indeed decreases electrons coming out from PSII. I wonder if O₂ can bind the non-heme iron in PSII upon bicarbonate dissociation in anoxic condition. If yes, can O₂^{•-} be measured?

Can we see differences at the non-heme iron by EPR spectroscopy similarly to what is done in Roach et al., 2013?

Regarding the TL experiments, the measure of the Q-band would be quite informative on the QA status. In addition, the use of nigerin will remove differences between the cultures in the proton gradient or the electric field that may influence the recombination event.

Generally, the manuscript improved however, I still think that there are no experimental proofs that cyclic electron flow within PSII via Cytb559 is operating in the studied conditions. I would remove this hypothesis.

Minor comments

Line 20 : "we" not "We".

Lines 200-202: I would remove the last three lines of Figure 2 description. I would rather move them to the supplementary Figure 4.

Line 419 : "glucose" not "Glucose" (please, correct both of them)

Line 427 : The ";" is not needed.

Point to point replies

Reviewer #2 (Remarks to the Author):

The authors have addressed our comments in great detail, with admirably open-minds, and much patience. Their balanced responses, additional experiments and positivity has resulted in major improvements in the article.

The authors chose to maintain the TL data against our suggestions (point 5): we fully agree with the authors decision. Our suggestion to remove all the TL was due to our (blinkered) view of the TL data as only being related to the cyclic electron flow hypothesis. Its utility in the present context is well justified.

We are happy to recommend publication.

A W Rutherford and A Fantuzzi

Reviewer #4 (Remarks to the Author):

In this manuscript, the authors aim at understanding the origin of the decrease of linear electron flow in photosynthesis in fluctuating light after anoxia conditions. The decrease of electrons flux in photosynthesis after successive exposures to light limits H₂ production, a potential renewable energy source. Experiments presented in the manuscript suggest that the decrease of H₂ production in the studied conditions depends on a drop in PSII activity. The mechanism that cause the decrease of PSII activity is investigated. In the revised manuscript, Milrad and co-workers address some of the questions previously raised.

General comments

1. New added experiments show that bicarbonate affects the PSII activity, reducing the electrons coming out from PSII. This is not specific of anoxic condition but it is an interesting result.
 - We thank the reviewer, indeed the original *in vitro* mechanistic description of the bicarbonate release mechanism in Brinkert *et. al.* was attributed to oxic conditions. However, in this paper we focused our view in anoxic condition, and to our understanding, added a new point of view on this interesting subject, alongside interesting impacts that this phenomenon generates.
2. Authors suggest that an over-reduction of the PQ-pool increases the formation of Q_A^{•-} that has the effect to displace the bicarbonate from the non-heme iron of PSII. The unoccupied site at the non-heme iron is taken by O₂, which in turn receives an electron by Q_A^{•-} forming O₂^{•-}. This mechanism indeed decreases electrons coming out from PSII. I wonder if O₂ can bind the non-heme iron in PSII upon bicarbonate dissociation in anoxic condition. If yes, can O₂^{•-} be measured?
 - We thank the reviewer for a very interesting question, which we do hope to understand in the future. As far as we can do, it is not feasible to measure anions using our instruments. As described in previous publications, our MIMS has a silicon membrane which enables only dissolved gas to penetrate toward the mass-spec. However, we would be more than pleased to see other working groups attempt such measurements and hope to inspire such research.
3. Can we see differences at the non-heme iron by EPR spectroscopy similarly to what is done in Roach et al., 2013?

- As far as we understand, and as a disclaimer, EPR and structural analysis are not our field of expertise, it could be possible to see changes in EPR between different PSII assembly structures (see Zabret et al., 2021). But so far, we do not know of any measurement which pin-pointed the non-heme iron in different states such as those which were proposed here.
4. Regarding the TL experiments, the measure of the Q-band would be quite informative on the Q_A status. In addition, the use of nigericin will remove differences between the cultures in the proton gradient or the electric field that may influence the recombination event.
 - We are grateful for this proposal, but since we decreased our focus on these measurements, we ended up taking down some of our measurements and only use these result in order to show that the exposure to light triggers changes in PSII's apparent activity, and that such changes relax back to their basic state following 15 minutes of darkness. Unfortunately, any further conclusions are unambiguous, and we would therefore refrain from conducting any further tests on the subject, in the scope of this work. Regarding future studies, we agree with this suggestion and we thank the reviewer for this proposal. Indeed, these so far 'open questions' would require more research and we hope to be able to answer them in the near future.
 5. Generally, the manuscript improved however, I still think that there are no experimental proofs that cyclic electron flow within PSII via Cytb559 is operating in the studied conditions. I would remove this hypothesis.
 - We thank the reviewer for the kind words and the efforts that we done in the process. We acknowledge the fact that reviewing other's research can be challenging and time-consuming. Admittedly, the efforts that were made assisted us in our process and generally the debate and discussion over the interpretation of our results was more than satisfactory for us. We are more than grateful for the productive comments and believe that indeed this manuscript was greatly improved thanks to these efforts. We could not have done it without it, and for that we are grateful. Concerning the remaining concerns, indeed we cannot claim that we proved the involvement of cyclic PSII, and specifically the Cytb₅₅₉ route, using our results. Instead, we do propose that O₂ reduction by the Q_A site governs the observed apparent quantum yield of PSII, and that this process is initiated by the release of the HCO₃⁻ molecule. Since we also observed that the process is not only triggered by the presence of dissolved CO₂, we felt that, an additional explanation must be presented. Therefore, we proposed two mechanisms that could comply with our results, based on suggestions that were done in previous research. One of these proposals involve cyclic PSII activation. We know that it is not solid, but still we feel that such hypothesis is not contradicted by our observations and therefore can be suggested. We added a disclaimer sentence to the revised manuscript which state: "However, the mechanism of such a route remains elusive and will require further studies in the future. In any case, the added electron pressure will be alleviated by an alternative local electron acceptor, which could possibly be O₂ in the proximity of PSII." We hope that this would be enough to ease these concerns and explain our meaning in this proposed alternative hypothesis.

Minor comments

Line 20 : “we” not “We”.

Lines 200-202: I would remove the last three lines of Figure 2 description. I would rather move them to the supplementary Figure 4.

Line 419 : “glucose” not “Glucose” (please, correct both of them)

Line 427 : The “;” is not needed.

- All minor comments were addressed and changed accordingly.